# Panning for Gold in Federated Learning: Targeted Text Extraction under Arbitrarily Large-Scale Aggregation

**Hong-Min Chu**[1]    **Jonas Geiping**[1]    **Liam Fowl**[1]    **Micah Goldblum**[2]    **Tom Goldstein**[1]
[1] University of Maryland    [2] New York University
{hmchu, jgeiping, lfowl, tomg}@umd.edu    goldblum@nyu.edu

## Abstract

As federated learning (FL) matures, privacy attacks against FL systems in turn become more numerous and complex. Attacks on language models have progressed from recovering single sentences in simple classification tasks to recovering larger parts of user data. Current attacks against federated language models are sequence-agnostic and aim to extract as much data as possible from an FL update - often at the expense of fidelity for any particular sequence. Because of this, current attacks fail to extract any meaningful data under large-scale aggregation. In realistic settings, an attacker cares most about a small portion of user data that contains sensitive personal information, for example sequences containing the phrase "my credit card number is ...". In this work, we propose the first attack on FL that achieves targeted extraction of sequences that contain privacy-critical phrases, whereby we employ maliciously modified parameters to allow the transformer itself to *filter* relevant sequences from aggregated user data and encode them in the gradient update. Our attack can effectively extract sequences of interest even against extremely large-scale aggregation.

## 1 Introduction

Industrial machine learning models are often trained on large sets of user data. In a traditional centralized training paradigm, this is done by aggregating user data into a large repository. Unfortunately, when user data contains personal information in the form of text, images, or other media, dataset aggregation leads to significant security, regulatory, and liability risks.

Against this backdrop, federated learning (FL) has emerged as a popular way to train models with *decentralized data*, that is without the need for a central party to host a dataset. By exchanging only model gradients, user devices collaboratively train a model without the direct exchange of plaintext data. In many applications, FL is slower than centralized training (Bonawitz et al., 2019), but the privacy benefits outweigh the costs, especially in next-word text prediction which requires training on private text from smartphones (Hard et al., 2019).

Privacy through federated learning is sometimes taken for granted. In reality, the actual privacy achieved by federated learning systems depends on a large number of factors and parameters – model size, architecture, number of users, the aggregation scheme, and more. Attacks against privacy in FL probe this boundary, empirically discovering pitfalls that should be considered and avoided when designing federated protocols (Phong et al., 2017; Melis et al., 2019; Geiping et al., 2020).

In this work, we study the security of federated learning systems involving transformer architectures (Vaswani et al., 2017) which form the backbone of many recent advancements in natural language processing (Brown et al., 2020; Dosovitskiy et al., 2021; Jumper et al., 2021), and especially applications in text, which represent a key point of interest in many modern applications of federated learning (Paulik et al., 2021; Dimitriadis et al., 2022). Our main threat model of interest is the *untrusted server* scenario, also known as the *malicious server* scenario, in which the server may make changes to model parameters in order to break user privacy. This is in contrast to the *honest-but-curious* threat model, in which no malicious changes are permitted to the model training protocol. *Untrusted server* scenarios are of critical importance from a user-centric privacy perspective.

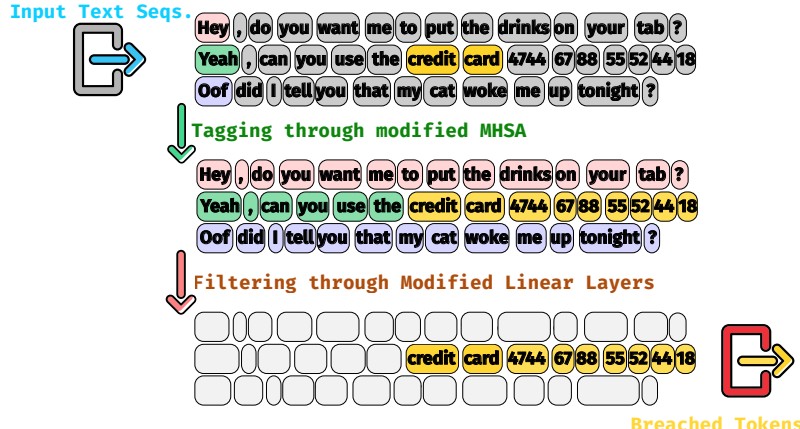

**Figure 1: The proposed attack "tags" and filters tokens so that they can be reconstructed from gradient information.** Malicious model parameters for a standard transformers (here with a causal structure) are sent to user devices. The attack uses one head to *tag* each token in a sequence with the first token of that sequence, (here in red, green, blue). This enables the attacker to group tokens into sequences after they are extracted from a gradient update. The attacker then uses two more heads to *tag* each token that follows the key words "credit" and "card" (yellow). These yellow target tokens will influence the gradient computation, while others will be *filtered* out. Finally the attacker recovers the targeted tokens from the gradient of the modified model returned by the user.

If a federated system is supposed to uphold privacy, then ideally this privacy can be guaranteed without having to assert full trust in the server. After all, a perfectly trustworthy server could run the simplest FL protocol: Centralize all user data, promise not to share it, train a centralized model, and delete all data. Another way to look at *untrusted server* threats is to view them as a glimpse of worst-case dataset security. Even if we believe a server will uphold privacy, we might wonder about the worst-case loss in privacy that would arise if this server is even briefly compromised (through either classical security breaches or poisoning attacks (Bagdasaryan et al., 2019)) and acts maliciously.

A major strategy available to a malicious server is to modify the current state of a machine learning model as it is being trained, and then broadcast this corrupted model to the users. As the model is directly executed on each user device, this can be considered an analogue to *untrusted code instructions* that are being evaluated on a user's private data (OWASP, 2022; Fowl et al., 2022).

Despite the inherent power that a malicious server has, extracting user data is still extremely difficult when gradients are aggregated over many users, in which case the averaged gradient does not contain enough entries to record the whole global training batch. For this reason, existing attacks on text models only recover user data in scenarios where the number of model parameters is significantly larger than the number of tokens in a user update (Fowl et al., 2022; Gupta et al., 2022; Dimitrov et al., 2022; Pasquini et al., 2021). In some cases, attacks can siphon random examples of user data, but only through a large number of repeated queries (Wen et al., 2022).

In this work, we discuss a novel attack on text models whereby a malicious server is able to pick and choose which data to encode and extract from the model gradient, even with industrial-scale aggregation. The attacker selects a *trigger* phrase, such as "credit card number" or "social security number," and extracts all tokens of user data that follow the occurrence of this trigger. We call this process, in which we sift selected phrases out of a large corpus of user data, "panning." In comparison to existing attacks, this attack does not degrade when very many user updates are securely aggregated (Bonawitz et al., 2017). For this reason, panning is an essential shift in capabilities for attacks against transformer-based models in federated learning.

## 2 BACKGROUND AND APPLICATION EXAMPLES

Text models were the first and most successful systems where federated learning has been used in industrial settings. These applications include keystroke prediction Hard et al. (2019); Ramaswamy et al. (2019), settings search Bonawitz et al. (2019), news personalization (Paulik et al., 2021), and improved messenger services on Android (Google, 2022). In the latter case, the documentation of

Google messages[1], describes federated learning being used to improve ML models related to Google messages, for example "smart reply" features. FL is active by default on all Android devices with this app. According to the documentation, user conversations are protected only through secure aggregation (Bonawitz et al., 2017), which "*can't reveal your conversations or content to Google or anyone else [...], grouping many similar adjustments together so that Google can't inspect an adjustment from a single device*". In this work, we argue that this system of protecting privacy leaves users open to targeted extraction attacks from a malicious server update sent to the user, provided the system uses a transformer-based machine learning model.

Known attacks against privacy in federated language models are not capable of breaking privacy in such large aggregation settings. The earliest attacks against transformers in Zhu et al. (2019b); Deng et al. (2021) operate in the "honest-but-curious" threat model. These attacks have resulted in only limited success, recovering data only from single sequences. In practice, "single sequences" imply that user updates are not securely aggregated, each individual user sends an individual update that represents only their data. Recent developments in Dimitrov et al. (2022) and Gupta et al. (2022) have pushed the envelope, reconstructing more and more text fragments, even from multiple sequences in this restricted threat model. On the other side, malicious server threat models, for example in Pasquini et al. (2021) and Fowl et al. (2022) can recover more information and identify or respectively reconstruct from aggregates of hundreds of sequences.

These attacks all attempt to recover all user data from an aggregated gradient, putting equal emphasis on the fidelity of each reconstructed sequence. This strategy inevitably decreases the capability of previous attacks to recover accurate individual sequences as the number of sequences and tokens increases, and eventually the extracted sequences become meaningless combinations of tokens.

Attacks that break large scale aggregation generally require additional assumptions. Lam et al. (2021) show that aggregation can be breached if side-channel metadata is available to the server, Pasquini et al. (2021) show that if users can be addressed separately by the server and each user only has limited data, malicious parameters with zero-gradients can be used to suppress all other users and break aggregation. This trick can also be used for targeted membership inference, checking if a user owns a specific datapoint. Further, Wen et al. (2022) shows that if malicious servers are allowed to send multiple queries, then they can "fish" for single data points, even from aggregated data.

Yet, in a real-life scenario, among all data a user computes and aggregates model updates with, often only a few of them contain information valuable to a potential attacker. For example, out of a 50-sentence online conversation between a user and a bank agent, an attacker may only be interested in the particular sentence that contains the user's social security number. This poses the question of whether it is possible for an attacker to dedicate the capacity of the gradient to only a limited number of target sequences with a specific set of keywords or triggers.

**Threat Model - Untrusted Server** Our threat model contains two parties. First, a user (or group of aggregated users) that owns text, which contains private information following a set of keywords $K = \{k_1, \ldots, k_n\}$. Second, a malicious server that aims to "pan", e.g., perform targeted extraction of user tokens that immediately follow one of these keywords. We assume that the number of aggregated sequences may be unlimited, but the number of sequences matching the combination of keywords is limited. We assume that the federated learning exchange is otherwise secure: Both parties agree beforehand on a transparent implementation for both model architecture and user-side protocol that is vetted by public examination. The only attack vector for the server is the model update sent indiscriminately to all users in the group. In this way, the presence of the attack may be non-obvious because it does not require deploying any new executable code, and it is embedded in network parameters that are seldom, if ever, inspected.

Even stronger threat models have also been investigated (Frey, 2021; Boenisch et al., 2021; Fowl et al., 2021). Threat models considered in Frey (2021) allow the server to execute arbitrary pieces of code on user devices, while Fowl et al. (2021); Boenisch et al. (2021) assumes a server can maliciously modify model architectures. We also note that the role of the attacker need not be played by the party that owns the server, and any individual/party that has access to the model update anywhere in the pipeline can assume the role. Particularly, a man-in-the-middle attacker can inject the attack into model parameters before broadcasting to users, retrieve the resulting unsecured user update, and replace the outgoing update with seemingly normal model gradients.

---

[1]https://support.google.com/messages/answer/932790

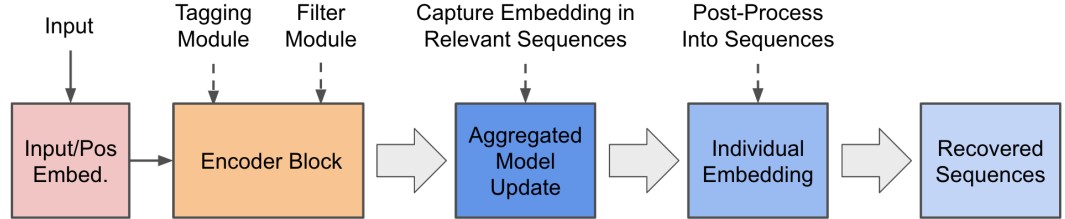

**Figure 2:** An overview of the mechanics of the proposed panning attack. The attacker reprograms the encoder block to tag and filter input sequences. This encodes keyword-specific information into the model gradient, which enables the attacker to "pan" for targeted tokens from even large aggregated user updates. The attacker can then post-process the tokens retrieved from the model gradient to recover original sequences.

**Remark** (Peer-to-Peer Attacks between Users.). *This threat model applies even if the server is fully benign: If in any round of training, a group of malicious users can control enough updates to replace the model parameters (Bagdasaryan et al., 2019; Bhagoji et al., 2019) in a single round of training, then the server will unwittingly distribute these malicious model parameters in the next round of training, collect model updates from other, vulnerable users, aggregate them and redistribute the updated model. Because an attack through "panning" is invariant to aggregation sizes, this means that the group of attackers can then discover data from vulnerable users in the next round of training, while the server remains unaware of any breach privacy.*

## 3 METHOD

Here we discuss the possibility of a"panning" attack that performs extraction of *target* sequences that contain a specific set of keywords. The attack we describe below is specific to transformer architectures and based on modifications to attention mechanisms and linear layers. The key ideas can be summarized as follows:

1. Modification of multi-head attention blocks (MHSA) to "tag" the tokens of a sequence that contains specific keywords, as in Section 3.2.
2. Modification of linear layers to detect the tag and trigger capture of token embeddings belonging to target sequences, while filtering out non-target sequences, see Section 3.3.
3. Post-processing the relevant embeddings into target sequences, as in Section 3.4.

An overview of our attack is presented in Figure 2. The discussed attack draws on a family of works describing "analytic" attacks against models in federated learning that we describe in Section 3.1.

### 3.1 PRELIMINARIES

As discussed in Phong et al. (2017); Geiping et al. (2020); Fowl et al. (2021), the gradient of a linear layer parameterized by $(\mathbf{W}, \mathbf{b})$ whose forward pass is $y = \mathbf{W}f + \mathbf{b}$ reveals information about its input $f$. In particular, for any given row $W_m$ and the related bias entry $b_m$, we have

$$\nabla_{W_m} L = \frac{\partial L}{\partial y_m} \cdot \nabla_{W_m} y_m = \frac{\partial L}{\partial y_m} \cdot f, \quad \frac{\partial L}{\partial b_m} = \frac{\partial L}{\partial y_m} \cdot \frac{\partial y_m}{\partial b_m} = \frac{\partial L}{\partial y_m} \tag{1}$$

where $L(y)$ denotes the downstream loss computed on the layer output $y$. An attacker can then simply reconstruct $f$ by computing $f = \nabla_{W_m} L \oslash \frac{\partial L}{\partial b_m}$, where $\oslash$ denotes entry-wise division as.

However, attackers cannot easily apply this trick to transformers in federated learning. Firstly, transformers add together token embeddings and positional embeddings, and have residual connections that add the output of multi-head self-attention (MHSA) to these mixed embeddings before applying the first fully-connected linear layer. In other words, the individual input $f_{i,j}$ of the first linear layer of the first encoder block is

$$f_{i,j} = e_{i,j} + p_j + \texttt{MHSA}(\{e_{i,j'} + p_{j'}\}_{j'=1}^{\ell}; \tag{2}$$

where $e_{i,j}$ is the embedding of the $j$-th token in the $i$-th sequence, $p_j$ is the $j$-th positional embedding, and $\texttt{MHSA}(\{e_{i,j'} + p_{j'}\}_{j'=1}^{\ell}, j)$ is the $j$-th entry after applying MHSA on sequence $i$. Secondly,

assuming the model update is computed over $B$ sequences of length $\ell$, an attacker only has access to the *average gradient* aggregated over all $B\ell$ tokens, The aggregated gradients of the first linear layer are instead

$$\sum_{i=1}^{B}\sum_{j=1}^{\ell}\nabla_{W_m}L_{i,j} = \sum_{i=1}^{B}\sum_{j=1}^{\ell}(\frac{\partial L}{\partial y_m} \cdot f_{i,j}), \quad \sum_{i=1}^{B}\sum_{j=1}^{\ell}\frac{\partial L_{i,j}}{\partial b_m} = \sum_{i=1}^{B}\sum_{j=1}^{\ell}\frac{\partial L_{i,j}}{\partial y_m}. \tag{3}$$

Recovering an individual $f_{i,j}$ from this gradient cocktail is non-trivial.

In summary, to recover a user's input sequences, an attacker needs to (1) isolate out individual mixed embedding $f_{i,j}$, (2) split the mixed embedding $f_{i,j}$ into a token embedding $e_{i,j}$ and position $p_{i,j}$ , and (3) match $e_{i,j}$ to the most probable word. If all tokens are uniformly represented in the gradient, then step (1) may be impossible. Below, we show how the network can be put into a state where the gradient's information capacity is sparingly used to encode targeted sequences.

> **Big Picture**: To extract data from a transformer model update, the attacker needs to first extract individual mixed embeddings from aggregated (batch averaged) gradients, group these into sentences, and then split the mixed embeddings into their position and token.

## 3.2 TAGGING TARGET SEQUENCES

In this section, we describe how an attacker could modify parameters in the attention block to "tag" the mixed embedding that belongs to a sequence that contains the selected keywords $K = \{k_1, \ldots, k_N\}$. We illustrate the modification in Figure 3. For simplicity, we start by detailing this construction with only one keyword $k$. We refer to a sequence that contains $k$ as either the "target" sequence or "relevant" sequence. Other sequences are called "non-targeted" or "irrelevant."

To facilitate the construction, we assume the norm of token embedding $e^{(k)}$ associated with the target word $k$ to be larger than the norm of all other token and positional embeddings. The server can easily meet this assumption as both embedding matrices are also part of the model parameters under their control. With this assumption, we design a self-attention mechanism inspired by Fowl et al. (2022) that tags the target sequences by imprinting $e^{(k)}$ into every mixed embedding in the target sequence. To be more specific, let $(\mathbf{W}_Q, \mathbf{b}_Q), (\mathbf{W}_K, \mathbf{b}_K), (\mathbf{W}_V, \mathbf{b}_V)$ be the weight matrices and biases of query, key, and value layers of MSHA in the first encoder block respectively. We set each of the parameters as follows,

$$\begin{cases} \mathbf{W}_Q = \mathbf{0}, & \mathbf{b}_Q = \alpha e^{(k)}, \\ \mathbf{W}_K = \mathbf{I}_d, & \mathbf{b}_K = \mathbf{0}, \\ \mathbf{W}_V = \mathbf{T}_{d'}, & \mathbf{b}_V = \mathbf{0}, \end{cases} \quad \mathbf{T}_{d'}[q,r] = \begin{cases} 1 & q = r, q \le d', \\ 0 & \text{otherwise.} \end{cases} \tag{4}$$

where $\alpha$ is a large positive value (e.g. $10^8$) and $\mathbf{T}_{d'}$ is the truncated identity matrix with pre-defined dimension $d'$. $\mathbf{W}_K$ and $\mathbf{b}_K$ form an identity operator, and thus the key token $\mathbf{K}$ (and others) pass through unaltered. $\mathbf{W}_Q$ and $\mathbf{b}_Q$ are designed so that the query vectors $Q$ are each identical copies of $e^{(k)}$. Finally, $\mathbf{W}_V$ and $\mathbf{b}_V$ keep only the first $d'$ entries of the input embedding, and set all other positions in each embedding to zero.

Let's evaluate the results of the MHSA computation based on this construction. If a length-$\ell$ sequence $\{e_j\}_{j=1}^{\ell}$ contains the target $k$ at position $j'$, then the attention weight $\mathtt{softmax}(\frac{\mathbf{QK}^{\top}}{\sqrt{d}})$ will be large, forcing every word in the sentence to attend to the target word. In fact, when $\alpha$ is large enough, attention to the target is near-absolute. Thus, the resulting mixed embedding after the MHSA is:

$$f_j = e_j + p_j + \mathbf{T}_{d'}(e^{(k)} + p_{j'}) \tag{5}$$

In plain language, the first $d'$ entries of the keyword embedding vector are added to the first $d'$ positions of every embedding in the sequence. We say that the sequence is now *tagged* with the keyword. For causal language models, applying the construction in Equation (10) on masked MHSA turns the keyword into a trigger, where the attention imprints $e^{(k)}$ to tokens that appear *after* the keyword in the target sequences. Coupling this tagging approach with a filter introduced later in Section 3.3, one can perform targeted extraction of *sub-sequences* that start with the keyword $k$. On the other hand,

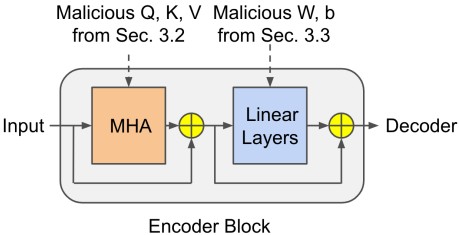

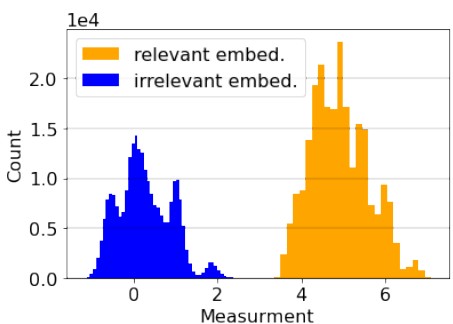

**Figure 3:** An illustration of our key modifications. We modify the query, key and value layer weights in the MHSA block to tag the mixed embedding of target sequences by content and position. We modify the feed-forward layer weights to bin tagged tokens into separate rows of the weight matrices $W$ and in this manner filter for only targeted tokens. $\oplus$ represents the add-and-norm operation in a transformer.

**Figure 4:** The measurement distributions of mixed embedding by $\mathbf{w} + \mathbf{w}'$ from Section 3.3 on GPT-2. The measurement distribution of irrelevant, non-target, embeddings (blue) exhibits clear separation from the measurement distribution of relevant target embeddings (orange). We set $\beta = 2.5$ here.

if the sequence is irrelevant, or in the case of causal MHSA, if the mixed embedding appears before $k$, then the imprinted embedding will be random from $e_1$ to $e_\ell$. This difference allows our malicious modification to tag the mixed embedding in target sequences by a keyword-specific imprint, and we will discuss in the next section how we filter out irrelevant sequences based on this imprint.

We note that extending the above construction to multiple keywords is straightforward. In particular, for $N$ keywords $k_1, \ldots, k_N$, one can simply repeat the above constructions for a separate head in the MHSA block for each keyword, and imprint each keyword-specific signature to different positions.

> **Big Picture**: The attention mechanism of a transformer is repurposed to attend strongly to the attacker's keywords, and add these keyword embeddings to all other entries in the sequences, thereby tagging them as important.

### 3.3 FILTERING NON-TARGETED TOKENS

Targeted tokens are those appearing in a relevant sequence containing a keyword. All other tokens in other sequences are not targets and should be filtered out. We want to remove them from the gradient aggregation to enable the attacker to isolate the few remaining target tokens in the gradient. We provide an illustration of the key idea in Figure 3. Our approach draws inspiration from Fowl et al. (2021), who performs gradient separation on aggregated gradients uncovered in Equation (3). We construct a malicious linear "imprint" layer by assigning to each row of $\mathbf{W}$ identical copies of a measurement vector $\mathbf{w}$. We then set the entries of the bias vector $\mathbf{b}$ to form an increasing sequence $b^m < b^{m+1}$. If the linear layer is followed by a ReLU activation, the aggregated gradient becomes

$$\sum_{i=1}^{B} \sum_{j=1}^{\ell} \nabla_{W_m} L_{i,j} = \sum_{i=1}^{B} \sum_{j=1}^{\ell} (g_m(f_{i,j}) \cdot \frac{\partial L}{\partial y_m} \cdot f_{i,j}), \ \sum_{i=1}^{B} \sum_{j=1}^{\ell} \frac{\partial L_{i,j}}{\partial b_m} = \sum_{i=1}^{B} \sum_{j=1}^{\ell} (g_m(f_{i,j}) \cdot \frac{\partial L_{i,j}}{\partial y_m}) \quad (6)$$

where $g_m(f_{i,j}) = 1$ if $\langle \mathbf{w}, f_{i,j} \rangle + b_m > 0$ and $g_m(f_{i,j}) = 0$ otherwise.

Intuitively, the ReLU outputs from this layer are like a thermometer for measuring the inner product $\langle \mathbf{w}, f_{i,j} > \rangle$. Because the bias entries are increasing, more ReLUs will be active if the value of $\mathbf{W} f_{i,j}$ is larger. The $m$-th ReLU turns on when $\langle \mathbf{w}, f_{i,j} > \rangle$ lies above the cutoff defined by $\mathbf{b}_m$.. The gradient for the row $W_m$ aggregates over all tokens with inner products above $\mathbf{b}_m$.

Now suppose that just one token $f'$ out of all $f_{i,j}$ uniquely satisfies $g_m(f') = 1$ and $g_{m+1}(f') = 0$. Then we can separate out the individual gradient for this token by computing

$$\sum_{i=1}^{B} \sum_{j=1}^{\ell} (\nabla_{W_m} L_{i,j} - \nabla_{W_{m+1}} L_{i,j}) = \frac{\partial L'}{\partial y_m} \cdot f', \quad \sum_{i=1}^{B} \sum_{j=1}^{\ell} (\frac{\partial L_{i,j}}{\partial b_m} - \frac{\partial L_{i,j}}{\partial b_{m+1}}) = \frac{\partial L'}{\partial y_m}. \quad (7)$$

We can now uniquely recover $f'$ just as we did from with a batch size of 1 in Equation (1). Therefore, if the attacker can accurately estimate the distribution of $\langle \mathbf{w} f_{i,j} \rangle$, they can then construct $\mathbf{b}$ accordingly to maximize the chance for every $f_{i,j}$ to fall uniquely in one of the bins $(b_m, b_{m+1})$.

It was noted by Fowl et al. (2022) that the distribution of $\langle w, (e_{i,j} + p_j) \rangle$ is approximately Gaussian if every entry of $\mathbf{w}$ is a random sample from a standard Gaussian. Assuming $\mathbf{b}$ has $M$ entries in total, we can optimize the attack by setting $b_m = -\Phi^{-1}(\frac{m}{M})$ where $\Phi^{-1}(\cdot)$ is the inverse of estimated Gaussian distribution. Under this choice, each bin $(b_m, b_{m+1})$ approximately contains the same probably mass. However, observe that the number of bins upper-bounds the capability of an attacker to recover user data; if too many samples pass through the bins will become over-saturated. Therefore, directly attempting to recover all $f_{i,j}$ becomes less effective against large-scale aggregation.

To address the issue, we propose a set of modifications for linear layers to pan for relevant mixed embeddings. Here, we describe how to program the first linear layer of the first encoder block, and discuss the extension to multiple encoder blocks in Appendix A. The key idea is to design $\mathbf{w}$ so that targeted tokens produce a large value of $\langle \mathbf{w}, f \rangle$ while untargeted tokens produce a small value. We then assign entries of $\mathbf{b}$ so that ReLUs only activate for tokens in the target distribution. To be more concrete, we scale the norm of $e^{(k)}$, the embedding of the given keyword, to $\beta > 1$, where $\beta$ controls the "signal strength" of the relevant mixed embedding, and helps to separate the irrelevant embedding as detailed later. We also sample the random measurement vector $\mathbf{w}$ from standard Gaussian and mask out the first $d'$ entries. Finally, we construct $\mathbf{w}'$ as

$$\mathbf{w}'[q] = \begin{cases} e^{(k)}[q], & q \leq \text{d'} \\ 0, & q > \text{d'}. \end{cases}$$

By setting each row of $\mathbf{W}$ to $(\mathbf{w} + \mathbf{w}')$, we have

$$\langle \mathbf{w} + \mathbf{w}', f_j \rangle = \langle \mathbf{w} + \mathbf{w}', e_j + p_j \rangle + \langle \mathbf{w}', \mathbf{T}_{d'}(e_{j'} + p_{j'}) \rangle, \tag{8}$$

by Eq. 5 where $j'$ is the position the MHSA attends to. Assuming $e_j$ and $p_j$ are both zero-mean (these can be chosen at will by the server), the measurement distribution of irrelevant embeddings remains zero-mean, while a mean shift of $(\beta \|\mathbf{w}'\|)^2$ can be observed in the distribution of the mixed target embeddings. We show the separation between both distributions empirically in Figure 4. The distinction allows us to adapt the binning strategy from Fowl et al. (2022) to focus on the distribution of relevant target embeddings. This allows the attack to condense all other embeddings into a small number of bins and dedicate the remaining bins to separating individual target embeddings.

> **Big Picture**: The attack leads to a bimodal distribution of embeddings. Targeted embeddings produce large activations in the linear imprint layer, while untargeted embeddings produce low activations.

## 3.4 POST-PROCESSING MIXED EMBEDDINGS INTO MEANINGFUL SEQUENCES

With individual mixed embeddings $f_{i,j}$ isolated into separate ReLU bins, we then need to extract the corresponding sequence index $i$, position index $j$ and the actual word from the vocabulary for each $f_{i,j}$. The tagging we describe above is used to label tokens based on the *keyword* that made them a target Section 3.2. We can also introduce a second tagging mechanism that imprints *positional information* onto each token.

To be more concrete, we construct a positional imprint using a second head in the MHSA block:

$$\begin{cases} \mathbf{W}_Q = \mathbf{O}, & \mathbf{b}_Q = \alpha p_0, \\ \mathbf{W}_K = \mathbf{I}_d, & \mathbf{b}_K = \mathbf{0}, \\ \mathbf{W}_V = \mathbf{T}_{d'}, & \mathbf{b}_V = \mathbf{0}, \end{cases} \quad \mathbf{T}_{d'}[q, r] = \begin{cases} 1 & q = r, d' < q \leq 2d', \\ 0 & \text{otherwise}. \end{cases}$$

For a given sequence $\{f_j\}_{j=1}^{\ell}$ with mixed-in embeddings $\{e_j\}_{j=1}^{\ell}$, the positional imprint outputs

$$f_j = e_j + p_j + \mathbf{T}_{d'}(e_1 + p_1), \tag{9}$$

where entries $d+1$ to $2d$ of the first token in every sequence are imprinted to mixed embedding in the same sequence. We then group $f_{i,j}$ into sequences with constrained K-means clustering (Bradley et al., 2000) using entries $d+1$ to $2d$ that encode sentence identity. Constrained $K$-means allows us to constrain the size of each cluster, which is no larger than sequence length $\ell$.

After grouping embeddings into sequences, the next step is to recover positions. As mixed embeddings in the same sequences are guaranteed to have distinct positions, we perform position recovery

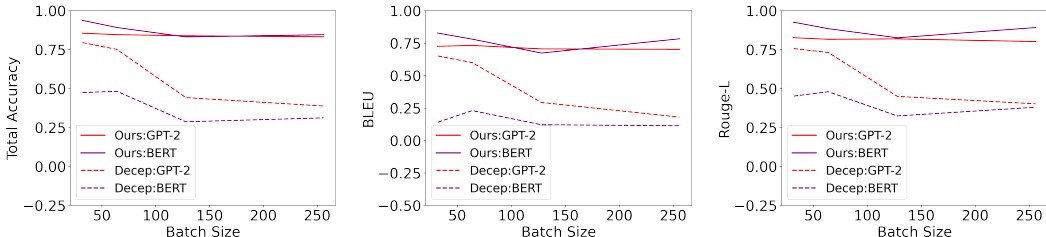

**Figure 5:** Comparison of our method with Decepticons, a recent Transformer-based attack, for different architectures across various batch sizes on targeted sequences. Decepticons is depicted with dashed lines. The sequence length is fixed to 32. Panning is a strictly stronger attack against sequences containing the target word.

on a sequence-by-sequence basis. For a set of mixed embeddings $\{f_{j'}\}$ grouped in the same sequence, identifying the most suitable unique position $j$ of each $f_{j'}$ can be viewed as a maximum weight bipartite matching problem, where we define the weight of a potential match $(f_{j'}, j)$ to be the correlation between $f_{j'}$ and $p_j$. We can solve the matching problem efficiently (Kuhn, 1955) and obtain the pure token embedding $e_{i,j}$ by subtracting the recovered positional embedding from $f_{i,j}$.

The final step is to associate each recovered token embedding $e_{i,j}$ with the actual words. (Fowl et al., 2022) discovered that the frequency of tokens presented in user updates can be estimated accurately. The estimated frequency dictates the maximum number of $e_{i,j}$ a specific word $v$ can be associated with. Then, taking the constraint into account, matching $e_{i,j}$ to the most suitable word $v$ can be formulated as another maximum weight bipartite matching problem, where the weight of a potential match $(e_{i,j}, v)$ is the correlation between $e_{i,j}$ and $e_v$.

> **Big Picture**: Individually recovered mixed embeddings (which are the sum of a position embedding and a token embedding) can be grouped by sequence. The tokens in a sequence can then be separated into token ID and position ID by solving bipartite matching problems.

## 4 EMPIRICAL EVALUATION OF THE ATTACK

In this section, we empirically evaluate our proposed attack on different transformer architectures commonly used in real-world applications. In particular, we consider the smallest variant of GPT-2 (Radford et al., 2019) with 124 million parameters, which is a causal language model with masked self-attention. We also consider a variant of BERT (Devlin et al., 2019) with 110 million parameters, which was used in previous works (Deng et al., 2021; Zhu et al., 2019a) for attack evaluation. Unlike GPT-2, BERT is trained as a masked language model which allows bidirectional self-attention.

We evaluate on the *wikitext* dataset (Merity et al., 2016), which we partition into articles. We simulate each user by sampling text from an article. We focus on fedSGD, where each user performs a single gradient step and sends a model update to the server. Note that many related protocols, such as federated averaging (McMahan et al., 2017), can be converted back to fedSGD, if the server is malicious and can control the number of local update steps or local batch size. We perform quantitative evaluations on our method with a range of different metrics. We measure success based on BLEU scores (Papineni et al., 2002), ROUGE-L (Lin, 2004), and total accuracy (where we only count exact matches of both token ID and positions ID). We provide additional details in Appendix A.

### 4.1 COMPARING WITH TRANSFORMER-BASED ATTACKS FOR MALICIOUS SERVERS

We begin by comparing our proposed method with Decepticons (Fowl et al., 2022), a recent transformer-based attack for the malicious server threat model, across various batch size and sequence lengths. To simulate the scenario where only a small number of sequences contain privacy-critical information, we select the first 3 sequences from each batch and replace one of the tokens of each sequence with a target keyword. For experiments on BERT, the replaced token is randomly selected. For experiments on GPT-2, we instead replace one of the first four tokens with a trigger keyword, which allows us to assess the impact of imperfect imprints created by masked self-attention. Due to causal language modeling, an attacker can only recover tokens *following* the trigger keywords for GPT-2. The quantitative results are then evaluated only on these target sequences.

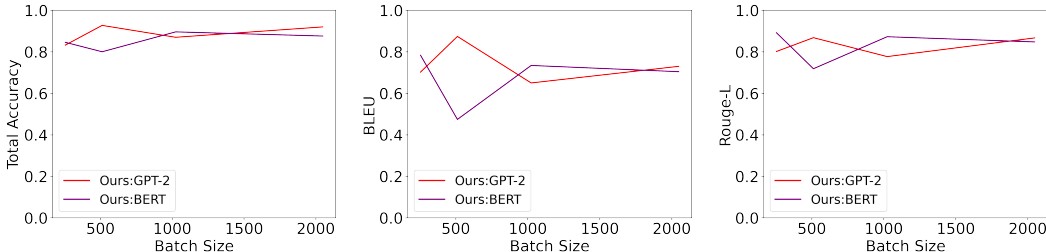

**Figure 6:** Experiment on our attack across different architectures for large batch size. The sequence length is fixed to 32. We evaluate our method only on the target sequences. *Panning* attack success remains constant as batch sizes increases to even large aggregation sizes.

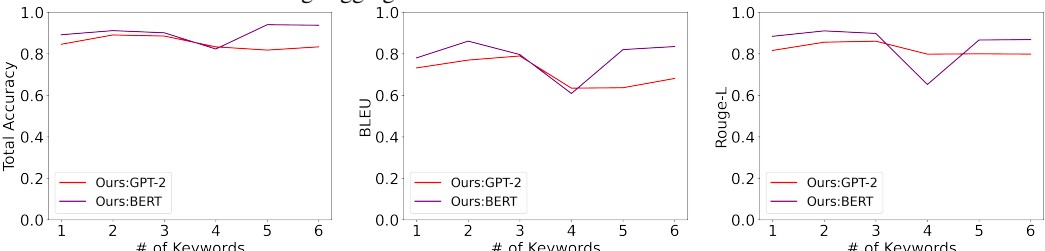

**Figure 7:** Experiment on our attack across different architectures for target phrases with different length. We fix the sequence length to 32 and batch size to 64 respectively. Our method is able to accurately capture relevant sequences given target phrases of various lengths.

We first fix the sequence length, and compare our method with Decepticons, while varying batch size in Figure 5. We see that the performance of Decepticons drops significantly as the batch size increases beyond 128, however the proposed panning attack mechanism is able to maintain a steady reconstruction quality on the targeted sequences as the batch size increases. In this sense, the new attack really is *independent of aggregation size*, and rather only depends on the number of keyword occurrences. We push this further in the next section.

## 4.2 Experiments with Large-Scale Aggregation and Multiple Keywords

Next, we evaluate our proposed methods under more extreme conditions. First, we assess the capability of our method to capture relevant sequences under extremely large-scale aggregation. We fix the sequence length to 32, and experiments with batch size up to 2048, and summarize the results in Figure 6. As demonstrated in the figure, the fidelity of our recovered relevant sequences remains high and is still largely agnostic to batch size. This observation validates the danger of the tagging and filtering strategy, which can capture tokens in targeted sequences even against extremely large-scale aggregation. We note that at this level, it becomes infeasible to even compare to Decepticons, due to the increased computational complexity to run the algorithm proposed therein for 65536 tokens.

Additionally, we investigate how the number of keywords $K$ impacts the performance of our attack. We perform experiments on $K \in \{1, 2, 3, 4, 5, 6\}$, We randomly replace a length-$K$ sub-sequence with a length-$K$ target phrase in each of the first 3 sentences. We present results in Figure 7, and demonstrate that our attack is able to recover sequences relevant to a various number of keywords. We note that both GPT-2 and BERT include 12 heads and allow us to target 11 keywords, which is enough to target most sensitive information in the real world.

## 5 Conclusion

In this paper, we describe a novel vulnerability for transformers used in a federated setting. This vulnerability opens the door to an attack through malicious parameter vectors that can "pan" for sequences that contain private information based on specified keywords. The attack allows an adversary to accurately capture sentences that contain the keywords out of aggregated updates from thousands of sequences. The attack injects malicious parameters into the transformer whereby sensitive sequences are "tagged" with a unique signature, and filtered to recover these targeted sequences. This attack reflects a notable shift in the capabilities of data reconstruction attacks in federated learning as the attack succeeds under seemingly arbitrary amounts of aggregation.

ETHICS AND MITIGATIONS STATEMENT

We describe an attack that significantly extends the capabilities of data extraction attacks for transformer models in federated learning. However, this attack only extends the capabilities of a malicious server and furthers our understanding that privacy cannot be guaranteed against such a server with only secure aggregation. Yet, in examples such as the message application discussed in Section 2, the described attack in this paper does *not* actually extend the capabilities of the server, as the federated learning system considered therein is implemented on a good-will basis without the ability for a user to inspect and vet the implementation of the protocol. From a security perspective, the server is so far not constrained to protocol, and is already capable of downloading user data - without federated learning. Both with and without our attack, the server in this example remains with current implementations only legally bounded to minimize their use of user data.

In terms of mitigation opportunities, two lines of defense algorithms are possible. The first line of defense algorithms involves (automatic) inspection of model parameters, and the second line of algorithms leverages differential privacy. We provide discussions for each separately.

**Mitigations through Inspection**    The vulnerability induced via malicious modifications of transformer parameters, as discussed in this work exhibits identifiable characteristics. Yet, there are several problems with relying on parameter inspection as a defense to this attack. First, awareness of this type of attack is necessary to know what to look for in terms of "conspicuous" parameters, which we hope to provide with this work.

Second, parameter inspection, at least on the server side, is not implemented in any major FL framework (see Bonawitz et al. (2021); Dimitriadis et al. (2022)). As such, the point of even being able to inspect parameters on a user side would require a user to have "rooted" their device - making this defense currently infeasible for the average user concerned about privacy.

Finally, inspection defenses naturally put users on the "back foot" in a back-and-forth between ever-adapting attackers and constantly defending users. Both parties need to agree about the implementation of all defenses, so that the defenses used on the user side are always known to the attacker, i.e. the server. On the other hand though, knowledge of attacks is limited until such vulnerabilities are published.

Because of the asymmetry induced by these sorts of defenses, we strongly advise against relying only on fixed sets of parameter inspection rules to avoid attacks like this.

**Mitigations through Differential Privacy**    We argue that strong user-level differential privacy, i.e. mechanisms applied directly on the user device to all outgoing model updates, remains the defense of choice against the attacks in the untrusted or malicious server threat model that we consider here. Common differential-privacy-based defense algorithms involve gradient noising and clipping, and we provide additional experiments to evaluate the effectiveness of these defenses Appendix C.

REPRODUCIBILITY STATEMENT

We perform all our experiments based on the official implementation provided in Fowl et al. (2022). We provide details of parameters and evaluations for our experiments, as well as pre-processing steps for the *wikitext* dataset in Appendix A.

ACKNOWLEDGMENTS

This work is supported by ONR MURI program, DARPA GARD (HR00112020007), and the National Science Foundation (IIS-2212182 and DMS-1912866). Further support was provided by Capital One Bank.

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

# A  ADDITIONAL DETAILS

In this section, we first describe how to reprogram multiple linear layers as hinted in Section 3.3. We then discuss variants of our construction for MHSA and linear layers that improve performance in practice.

## A.1  EXTENSION TO MULTIPLE LINEAR LAYERS

Linear layers in encoder blocks other than the first can provide more bins for a finer-grain gradient separation. Assuming we have $H$ encoder blocks, We can apply the same construction technique for every $\mathbf{W}^{(h)}$ of the first linear layer of the $h$-th encoder block. For bias, we instead compute $b_m^{(h)} = -\Phi^{-1}(\frac{Hh+m}{HM})$, where we essentially concatenate all the bias vectors, assign bin thresholds on the concatenated bias with $HM$ entries, and separate it back to corresponding linear layers.

Additionally, we need to disable MHSA blocks other than the first by setting the output to zero, and also partially disable the feed-forward block of each encoder block by setting all output entries except for the last to zero. Note that the output of feed-forward cannot be entirely disabled, or the gradient to the first linear layer will become zero. These additional treatments allow inputs of each feed-forward block to stay the same except for the last entry through residual connection.

## A.2  ALTERNATIVE CONSTRUCTION OF MHSA BLOCK AND LINEAR LAYER

Here we provide an alternative construction of the MHSA block. In particular, we instead use the following,

$$\begin{cases} \mathbf{W}_Q = \mathbf{0}, & \mathbf{b}_Q = \alpha e^{(k)}, \\ \mathbf{W}_K = \mathbf{I}_d, & \mathbf{b}_K = \mathbf{0}, \\ \mathbf{W}_V = \mathbf{T}_{d'}, & \mathbf{b}_V = \mathbf{0}, \end{cases} \quad \mathbf{T}_{d'}[q,r] = \begin{cases} 1 & q+d' = r, q \leq d', \\ 0 & \text{otherwise.} \end{cases} \tag{10}$$

where the key difference is that the new $\mathbf{T}_{d'}$ is instead a truncated shift matrix that shifts the entries $d'+1$ to $2d'$ of the input to the first $d'$ entries. By additionally setting the first $d'$ entries of each token embedding and position embedding to be zero, the resulting mixed embedding $f_{i,j}$ from Equation (2) will have their first $d'$ entries contain only the imprint information. On the other hand, the original $f_{i,j}$ whose first $d'$ entries are a mixture of imprint and original token/positional embedding, To apply this construction we also need set $\mathbf{w}'$ in Section 3.3 as $\mathbf{w}' = \mathbf{T}_{d'}e^{(k)}$ instead. Comparing to the original construction, this alternative provides better performance in practice.

# B  ADDITIONAL EXPERIMENT DETAILS

In this section we provide additional details for our experiments. We start by detailing hyperparameters. We set $\alpha = 10^{12}$ for GPT-2 and $\alpha = 10^8$ for BERT across all experiments. For GPT-2, we set $\beta = 10$ for GPT-2 if batch size is smaller than 256 and $\beta = 3$ for GPT-2 if otherwise. For BERT, we set $\beta = 10$ for all single keyword experiments, and we set $\beta = \frac{5}{K}$ for experiments on multiple keywords, where $K$ is the number of keywords. We set $d'$, the block size to be truncated and shifted as described in Section 3.2, to 32.

Next, we describe our procedure to process articles into sequences of tokens for each user. For experiments on GPT-2, the articles are tokenized with GPT-2 (BPE) tokenizer. For experiments on BERT, we tokenize the articles with the original BERT tokenizer. Then, given a sequence length $\ell$, we concatenate all the words in an article into one array, and partition the array into sequences of length $\ell$. The left-over words are discarded. Then, given the batch size $B$, we keep the first $B$ sequences for each user and discard the rest. The user then computes their model update based on the remaining $B$ sequences.

For each quantitative metric, the reported result is obtained over the average over the first 10 users with enough data. That is, for a given combination of batch size and sequence length, a user only contributes to the final result if his/her data is enough to fill in the required number of tokens.

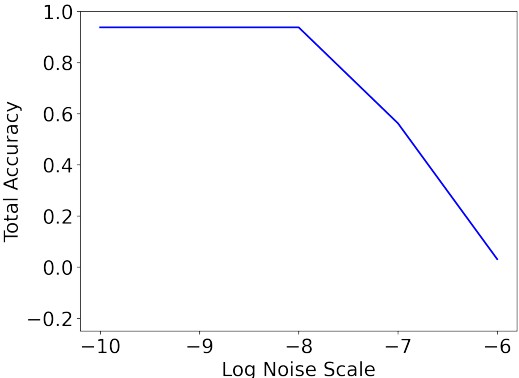

**Figure 8:** Experiments on 3-layer transformer with different noise scale. The x-axis represents the value after taking base-10 log on the noise scale.

**Discussion about FedAVG:**   In this work we focus on fedSGD as a template for federated learning. However, this restriction is not as limiting as it might seem: A malicious server can side-step the restriction of FedAVG if the hyperparameters of the FedAVG protocol can be configured so that either the user runs only a single local update step or runs the update on a large enough batch size. Both variants effectively turn FedAVG into FedSGD.

## C   ADDITIONAL EXPERIMENTS FOR DIFFERENTIAL PRIVACY

Here we provide experiments where a user additionally adds Gaussian noise and performs gradient clipping before sending the model update to server. Notice that the presence of noise makes the measurement by Eqn. 7 noisy. Therefore, we only perform embedding recovery using Equation (7) when the magnitude of difference between bias gradient is larger than a certain threshold. The experiment is performed on a 3-layer transformer as ribed in (Wang et al., 2021), where we clip gradient norm to 1 and compare performance on different noise scales from $\{1e^{-6}, 1e^{-7}, 1e^{-8}, 1e^{-9}, 1e^{-10}\}$, and fix the threshold to be $1e^{-6}$. The experiment also uses batch size of 2 and sequence length of 32, where one of the sequences contains the target word. The experiment results are presented in Figure 8.

The results show that the attacker is still able to succeed when the noise scale is relatively small, but predictably fails if the gradient becomes sufficiently noisy. We also note that defense based on noisy gradient is more likely to succeed against attack on larger transformer models due to the inherently smaller gradient magnitude.

Overall, as promised by theory, applying stronger differential privacy eventually enforces user's privacy, and we believe related defense algorithms are more practically attractive. However, it is also known that stronger differential privacy degrades the utility of the final model. Finding a good balance between privacy and utility remains an important research direction.

## D   ADDITIONAL EXPERIMENTS

To verify that the attack is stable across sequence lengths, we run a series of additional experiments, where we fix the batch size and compare across sequence length in Figure 9. The figure shows that our proposed attack also outperforms Decepticons across all combinations of model and sequence length (although the performance of Decepticons is less sensitive to the change of sequence length).

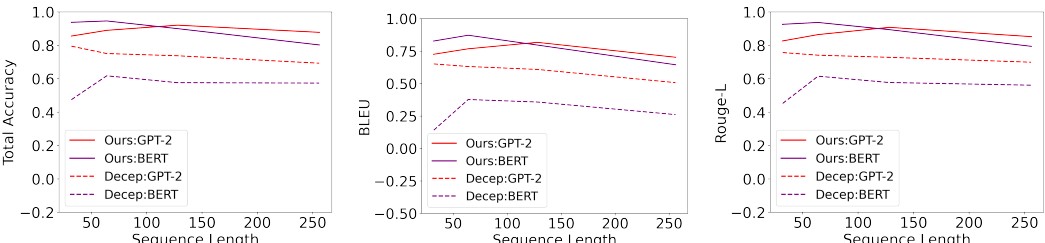

**Figure 9:** Comparison of our method with Decepticons for different architectures across various sequence lengths. The batch size is fixed to 32. We evaluate both methods only on the target sequences. Our attack remains agnostic to sequence length, and outperforms Decepticons in all metrics on target sequences.

