# OpenReview forum: "Panning for Gold in Federated Learning: Targeted Text Extraction under Arbitrarily Large-Scale Aggregation"
_ICLR.cc/2023/Conference — ICLR 2023 poster_

### Official Review · Reviewer_q4yB · 2022-10-15

**Confidence:** 3
**Correctness:** 2
**Technical Novelty And Significance:** 3
**Empirical Novelty And Significance:** 3
**Recommendation:** 5

**Clarity, Quality, Novelty And Reproducibility:**

Clarity: The clarity of this paper is poor. It is hard to understand for readers without sufficient knowledge of (Fowl et al. 2022).

Novelty: The paper is novel. It identifies a practical and novel attack setting, i.e. malicious server and tokens following specific keywords.

Quality: It is not very clear how the proposed method can be put in practice. Thus, I am doubtful about the quality of this paper.

Of course, as I am not very familiar with (Fowl et al. 2022), a central related work of this paper, I may make misunderstandings. Please correct me if there are any.

**Strength And Weaknesses:**

Strength:
- The problem setting studied in this paper is unique and practical. This paper studies malicious servers instead of honest-but-curious servers studied in existing works. This makes sense as one can never assume that an adversary is 'honest'. Also, the paper studies how to extract tokens following specific key words. This is also practical, since most sentences typed by users are not highly related to privacy.

Weaknesses:
- This paper does not seem self-contained and hard to understand without sufficient background. This paper heavily refers to related works (especially (Fowl. et al. 2022)) in the designs and equations. However, the intuition/implications of them are not sufficiently introduced. I list several examples here. (It is suggested that the authors number the equations).
    - Eqn. 1 is derived from Phong et al. (2017). However, although I knew the related work and understood it, it is not immediate for those who are unfamiliar with the related work. Also, it seems that $\frac{\partial b}{\partial L}$ should be $\frac{\partial L}{\partial b}$.
    - The equation following Eqn. 1 (Page 4) from Fowl et al. 2022 is also not immediate and hard to understand. Also, it seems that the equation only works for one layer of transformer block (otherwise, the input should not be token embeddings $e$ but output features from the previous layer).
    - The first equation on Page 6 from Fowl et al. 2022 is hard to understand without interpretation. The authors should interpret what the equation intuitively means. (e.g. organizing $w^Tx_j$ into bins).
    - The equation above Eqn. 5 from Fowl et al. 2022 is also hard to understand. I know it is more or less similar to Eqn. 2, but as they serve different purposes, it is still beneficial to explain it.
    - Thus, the paper does not seem self-contained. This paper is hard to understand without sufficient knowledge on (Fowl et al. 2022).

- It is not sure that the proposed method can be used in practice. I list several doubts on the applicability.
    - First, it is not sure that the proposed method is compatible with model training. If the local model is training smoothly (local loss decreasing) but the global model is not training (e.g. loss significantly increases after server aggregation), clients will not have the motivation to participate.
    - Second, the proposed method leads to non-negligible and weird parameter distribution. For example, a client can simply compute the norm of token embeddings and decide that the server model is weird and potentially malicious. By comparison, backdoor attacks on FL requires that the model outputs normally on normal data (Bhagoji et al. 2019, Bagdasaryan et al. 2019).
    - Third, for the motivating example of credit card numbers, the sensitive information (i.e. the card number) is a string of numbers which is almost certainly out-of-vocabulary. Therefore, the card number may not even have a token embedding, making the retrieval impossible.
    - Finally, the experiments use FedSGD (i.e. communication every iteration) as the attacked method. However, FedSGD is rarely used in practice for its high communication cost. In practice, FedAvg, which averages the gradients for multiple local steps (e.g. ~100) is used. It is not sure how this method works with FedAvg. Furthermore, if secure aggregation is used, the server only obtains aggregated gradients. Then, it is not clear how the retrieved information can be traced back to specific users.
    - Of course, as I am not very familiar with (Fowl et al. 2022), a central related work of this paper, I may make misunderstandings. Please correct me if there are any.

**Summary Of The Paper:**

This paper studies the problem of obtaining private text data from federated transformer models. This paper studies the malicious server setting, where the server can arbitrarily tamper with the model sent to clients. This paper also studies a unique attack setting --- it focuses only on several privacy sensitive key words and the tokens following the key words, such as 'card number'. The proposed method elaborately designs transformer parameters such that the gradients directly reveal the token embeddings following the key words. Experiments are performed on real-world transformer models, where the proposed method outperforms a state-of-the-art baseline, Decepticon (Fowl et al. 2022).

**Summary Of The Review:**

From my perspective, the main problems of this paper are i) Not sufficiently self-contained; 2) Hard to be deployed in practice. Therefore, although this paper studies a practical and unique problem setting, I cannot recommend acceptance at this stage.

---

> ### Author Response · Authors · 2022-11-18
> **Response to Reviewer q4yB**
>
> We appreciate the reviewer’s recognition that the problem setting is novel and practical. Here we provide detailed responses to the reviewer's individual questions.
>
> **Paper presentation not self-contained:**
>
> We have made **significant revision** to the paper to include more explanations and intuitions, with particular emphasis on the equations that the reviewer mentioned in the response. We hope that the revision addresses the reviewer’s concerns to make the presentation easier to understand, and please do not hesitate to let us know if there are any further questions.
> We also note that the equation following Eqn. 1 (which is now Eqn. 2 in the revised paper) is indeed the breakdown of the gradient for the first linear layer in the first self-attention block. The subsequent self-attention blocks are disabled by setting their output to zeros to keep the mixed embedding unchanged. We also disable the output of each feed-forward block except for the last entry of the output to allow gradient backpropagation. We now include these details in the appendix of the updated paper.
>
> **Compatibility with model training:**
>
> We note that only a single update is required to extract user data via attacks as proposed in this work.
> Therefore, the server can perform a “one-shot” attack that asks users (or only a small subset!) to calculate their local update on a malicious update, but can continue to send honest updates afterward. The server further needs not to include this malicious update into the global model, so that the global model can be improved further.
> Further, in most applications of federated learning, models are continuously trained and re-trained on a federated system and only deployed afterward after careful vetting and validation. From a user perspective, this makes it impossible to pinpoint, whether a new model checkpoint sent by the server is a new model early in training, or a malicious one.
>
> **Different parameter distribution:**
>
> We agree that the malicious modifications of transformer parameters proposed in the paper exhibit identifiable characteristics. To some extent, this is why we publish this attack and want to increase awareness in the community for these kinds of new attacks.
>
> Yet, while defending against a specific, known attack through inspection would be simple, this starts a cat-and-mouse game between attacker and defender. For example, knowing that the client may perform norm inspection, the server can artificially modify the norms of the irrelevant token embeddings while still keeping the attack successful as long as the norm of the target embedding remains larger. In general, the server can often adapt the attack to bypass an existing, fixed inspection-based defense, as we also discuss in more detail in the general response.
>
> **String of numbers being out-of-vocabulary:**
>
> In general, modern tokenizers split a string of numbers into sub-word tokens that are in-vocabulary. Our attack is specifically designed to capture this sequence of tokens following a keyword, for example a credit card number such as 4744002453231234543 will be tokenized as ['47', '44', '00', '24', '53', '23', '123', '45', '43'] with the BPE tokenizer used in GPT-2.
>
> We also provide a qualitative example of reconstructions where the critical information contains a string of numbers.
> In this example, the user data contains 64 sequences, each with length 32, and the transformer architecture is GPT-2. We partially replace one of the sequences starting from the first token with the following sentences:
> > “My credit card number is 1234567. This is a privacy leaking sentence. Can we recover sensitive information?”
>
> Using “credit card number” as a key phrase, here’s the reconstruction given by our attack:
>
> > “**My credit card number** Art **1234567** Fair **This is** 1973 **privacy leaking sentence. Can we recover sensitive information**?“
>
> Bold texts represent exact matches on both tokens and positions.
> We find here that an attack through panning can target the sequence containing these key words and reconstruct text that captures sensitive information, even when the information itself is a string of numbers.

---

> > ### Author Response · Authors · 2022-11-18
> > **Response to Reviewer q4yB (cont.)**
> >
> > **FedAvg and secure aggregation:**
> >
> > We agree that FedAvg is more commonly used. In this submission we focus on fedSGD as a template for federated learning. However, this restriction is not as limiting as it might seem: A malicious server can side-step the restriction of FedAVG if the hyperparameters of the FedAVG protocol can be configured so that either the user runs only a single local update step or runs the update on a large enough batch size. Both variants effectively turn FedAVG into FedSGD.
> >
> > Furthermore, we do agree that if the gradients are aggregated across multiple users, the attacker cannot relate a specific piece of leaked information to a specific user. While secure aggregation is breached it still reduces to a secure shuffle of user data.
> > However, we note that oftentimes the attacker may only care about sensitive information itself, for example getting as many valid social security numbers as possible. Also, the recovered text might contain personally identifiable information by itself, which could simply contain, e.g. the name of the user in proximity to their social security number.
> >
> > Overall, we hope to have addressed your concerns through our substantial edits to the draft and additional clarification in this response. Please let us know if there are other things you’d like us to clarify or improve. If the changes and explanations made here sound satisfying to you, we would also appreciate it if you would think about raising your score in response.

---

> > > ### Comment · Reviewer_q4yB · 2022-11-18
> > > **A quick response to the 'malicious capability' of server.**
> > >
> > > I have a quick response to the authors regarding the 'malicious capability' of the server. **I think that although the malicious server setting is interesting, the authors gave the server inadequate capability that is enough to reduce the problem to a naive query search.**
> > >
> > > By assuming the server is 'malicious', the authors allow the server to do basically anything, e.g. encode the backdoor in the training program while doesn't allowing clients to inspect, arbitrarily setting the parameters of FedAvg, etc. However, if the 'malicious capability' of the server is really that strong, it is possible for the server simply add a backdoor to do the following.
> > >
> > > - Let the clients do a keyword query, e.g. "credit card number".
> > > - Return all the query results to the server.
> > >
> > > This seems even more straightforward and effective than doing all these stuff, given that the server is 'malicious' and can do anything (as you said in the response, detecting it would require users to root their device). Is my understanding correct? If it is not, please justify why a malicious server cannot do this, and why can the malicious server do things like
> > > - Writing a FL framework without allowing clients (and the public) to inspect.
> > > - Arbitrarily setting the training configuration parameter

---

> > > > ### Author Response · Authors · 2022-11-18
> > > > **Malicious Capability - We assume a realistic threat model**
> > > >
> > > > We appreciate the reviewer's further engagement to the discussion, and we hope our reply clarifies the questions raised by the reviewer.
> > > >
> > > > We believe our threat model is realistic. In particular, our threat model includes the condition that “Both parties agree beforehand on a transparent implementation for both model architecture and user-side protocol that is vetted by public examination.“, which is described in Sec.2.
> > > >
> > > > To be more specific, we agree with the reviewer that the scenario where the server can deploy arbitrary code on user devices is not as interesting. This is why we restrict the server to a given federated learning protocol, the implementation of which can be verified and open-source.
> > > > The only attack vector for the server is the update sent from the server to user devices to interact with this fixed implementation. This update package (described for example in Paulik et al, 2021) contains the model checkpoint that a user device is supposed to evaluate and the hyperparameter configuration with which to evaluate it. For example, Apple (as in Paulik et al.) has no a-priori knowledge of what learning rate and local batch size are appropriate to use when training a new FL model, so these hyperparameters have to be modified and tested. Yet, the main protocol remains untouched and only model updates are returned by the user device.
> > > >
> > > > In these systems deployed in practice, the implementation of the protocol is open-source and inspectable by all parties. On the other hand, model parameters and hyperparameter configuration would be continuously communicated and really are the inputs to the system, and as such
> > > >
> > > > Please refer to our threat model description for more details.

---

> > > > > ### Comment · Reviewer_q4yB · 2022-11-19
> > > > > **Thanks for the clarification. Apologies for my misunderstanding.**
> > > > >
> > > > > I apologize for my misunderstanding and thank the authors for clarification. I now understand that the attack can be performed with a publicly accessible FL framework.
> > > > >
> > > > > However, from my perspective, there are still two points that I cannot fully agree with the authors.
> > > > > - **Clients should have the ability to inspect model parameters**. As the framework implementation is public, it makes no sense to assume that users cannot inspect model parameters.
> > > > > - **FedAvg cannot be simply reduced to FedSGD**. Given that the framework implementation is public, I think the widely used FedAvg (i.e. clients perform many local steps) should be default and widely accepted.
> > > > >
> > > > > Nonetheless, I agree that the authors' clarifications bring more sense to the attack.

---

> > > > > > ### Author Response · Authors · 2022-11-21
> > > > > > **Our attack justifies incorporating related security measures in real systems**
> > > > > >
> > > > > > At this point, we are wondering if you are addressing us or the reference implementations that we quote?
> > > > > > We completely agree that implementations that ensure both of these steps would make FL safer (although the first would still be non-trivial to automate) against the attack we describe here. And we want to emphasize again that we believe part of the value of our work is to show that including these two steps in the defense systems is necessary.
> > > > > >
> > > > > > Taking a step back, we believe the publication of this new attack mechanism is important to the discussion of how to defend against it. Even if the attack is easy to defend against, a user first needs to be aware that this kind of attack is possible.
> > > > > >
> > > > > > With no awareness of this attack,
> > > > > > * How would a company understand why sufficient DP should be applied, on top of the aggregation used so far?
> > > > > > * How would someone auditing parameter vectors determine if a specific update package from server is malicious?
> > > > > > * How would someone know to limit the minimal number of local update steps in federated averaging?
> > > > > >
> > > > > > From a broader perspective, we believe our paper provides the justification for these steps that you have suggested after engaging with our paper. In hindsight, we both agree with these suggestions. But it is this hindsight that we aim to provide to the community with our submission.

---

> > > > > > > ### Comment · Reviewer_q4yB · 2022-11-22
> > > > > > > **Response**
> > > > > > >
> > > > > > > I would like to point out that the two points are about the unrealistic assumptions made by the authors.
> > > > > > >
> > > > > > > The authors state that existing implementations do not allow clients to inspect the received parameters, which I cannot agree. First, federated learning is not intended only to mobile devices. There are also cross-silo FL (see https://arxiv.org/abs/2107.06917), and in this case clients can (and should, due to transparency) check the received parameter vectors.
> > > > > > >
> > > > > > > I do agree that, without knowledge of this attack, users cannot be certain that it is really used. However, it is very possible that users would raise suspicion after seeing that, e.g. word embeddings have unit norm except for several words, $W_Q = 0, W_K=I$, as normal transformers never look this way.
> > > > > > >
> > > > > > > The authors state that FedAvg can be simply reduced to FedSGD because the server can arbitrarily choose the number of local updates, which I cannot agree. It is against common practice to try FedSGD for FL as FedAvg (and running local steps) has been the de facto standard. Doing so would be immediately suspicious.
> > > > > > >
> > > > > > > Finally, DP is also a common practice in large corporations like Google (and even in FL, see https://github.com/google/rappor, https://ai.googleblog.com/2022/02/federated-learning-with-formal.html). It is not unusual for companies to apply DP over FL. Although the numbers ($\sigma=1e-6$) cannot show that the proposed attack can be completely defended by DP, assuming that DP is not used in practical FL is not practical either.
> > > > > > >
> > > > > > > Above are my reply (and unaddressed concerns to this work).

---

> > > > > > > > ### Author Response · Authors · 2022-11-23
> > > > > > > > **Further Details based on FL Literature**
> > > > > > > >
> > > > > > > > First off, we're happy that you found the revised and improved presentation helpful.
> > > > > > > >
> > > > > > > >
> > > > > > > > Further we are happy to continue to discuss and clarify remaining questions.
> > > > > > > >
> > > > > > > > > There are also cross-silo FL (see https://arxiv.org/abs/2107.06917), and in this case clients can (and should, due to transparency) check the received parameter vectors.
> > > > > > > >
> > > > > > > > We agree, but what would clients look for, after receiving a stream of several thousand or several million floating point values, possibly multiples times per minute in a cross-silo application. This paper shows *a first example* how attacks could have been encoded into an update, *informing any such audits*.
> > > > > > > >
> > > > > > > > > However, it is very possible that users would raise suspicion after seeing that, e.g. word embeddings have unit norm
> > > > > > > >
> > > > > > > > User might as well think that these are specially initalized or pretrained models sent by the central server. Without knowledge that attacks like this are even conceivable, there is no assumption of maliciousness.
> > > > > > > >
> > > > > > > > > It is against common practice to try FedSGD for FL as FedAvg (and running local steps) has been the de facto standard.
> > > > > > > >
> > > > > > > > The literature is actually much more nuanced on this question. To keep our presentation factual, here we quote relevant passages from a recent well-received survey, Kairouz et al, "Advances and Open Problems in Federated Learning", 2021. Sec. 3.2.1 therein discusses this question and notes:
> > > > > > > > > These result show that if the number of local steps $K$ is smaller than $T /M^3$ then the (optimal) statistical term is dominating the rate. However, for typical cross-device applications we might have $T = 10^6$ and $M = 100$ (Table 2), implying $K = 1$.
> > > > > > > >
> > > > > > > > > Li et al. [303] directly analyzes the Federated Averaging algorithm, which applies $K$ steps of local updates on randomly sampled M clients in each round, and presents a rate that suggests local updates ($K > 1$) could slow down the convergence. Clarifying the regimes where $K > 1$ may hurt or help convergence is an important open problem.
> > > > > > > >
> > > > > > > > > How then to choose $K$? Performing more local updates at the clients will increase the divergence between the resulting local models at the clients, before they are averaged. As a result, the error convergence in terms of training loss versus the total number of sequential SGD steps $T K$ is slower. However, performing more local updates saves significant communication cost and reduces the time spent per iteration. The optimal number of local updates strikes a balance between these two phenomena and achieves the fastest error versus wallclock time convergence.
> > > > > > > >
> > > > > > > > Crucially, we don't want to summarize here that FedAvg is never used, but we do want to reiterate that the choice of updates steps $K$ has so far chiefly been an optimization question, not a security question. *Our work challenges this notion*.
> > > > > > > >
> > > > > > > > > Finally, DP is also a common practice in large corporations like Google (and even in FL, see https://github.com/google/rappor, https://ai.googleblog.com/2022/02/federated-learning-with-formal.html).
> > > > > > > >
> > > > > > > > This is not the conclusion that the linked article makes. Google is certainly heavily researching DP, but the article concludes that
> > > > > > > > *"We are still far from being able to say this approach is possible (let alone practical) for most ML models or product applications".*
> > > > > > > > We believe our results can strengthen the commitment to this type of research.

---

> > > > > > > > > ### Comment · Reviewer_q4yB · 2022-11-24
> > > > > > > > > **The reply makes sense. I am willing to raise my point.**
> > > > > > > > >
> > > > > > > > > Dear authors,
> > > > > > > > >
> > > > > > > > > I would like to apologize for jumping into some quick conclusions. I think your answers at least partially address my questions, although not all. I appreciate your efforts.
> > > > > > > > >
> > > > > > > > > I am willing to raise from 3 to 5 as you have shown the value of your work to some extent.

---

> > > ### Comment · Reviewer_q4yB · 2022-11-18
> > > **On the ability of the proposed attack to bypass defenses.**
> > >
> > > I also have a quick comment on the ability of the proposed attack to bypass defenses.
> > >
> > > In general, even when some participant in a learning system is malicious and want to do some attack, the attack should be negligible and not easily bypassed. This principle holds for inference-time adversarial attacks, where the noise added should be minimal and not identifiable by humans. Also, this principle holds for backdoor attacks, where the poisoned data should have only a small patch of "backdoor trigger", and the generated model should have normal outputs and performance on clean testing data.
> > >
> > > However, the proposed attack is against such principle. The proposed attack generates weird parameter distributions, and is not compatible with model training (as the authors have acknowledged this in the response). Therefore, I consider the inability of the proposed attack to bypass simple defenses as a serious weakness of the proposed method.
> > >
> > > I also thank the authors for providing background knowledge on GPT-2 tokenizers and how it deals with string of numbers.

---

> > > > ### Author Response · Authors · 2022-11-18
> > > > **The Attack bypasses existing defenses**
> > > >
> > > > We appreciate the reviewer's further engagement to the discussion, and we hope our reply clarifies the questions raised by the reviewer.
> > > >
> > > > In general, we believe that a paper needs not describe an undetectable attack to be worth the community’s attention. Instead, a paper already brings value to the community if the described attack breaks existing security measures and advances the community's understanding of threats a defense needs to cover. That is, the publication of new attacks informs the defender to include appropriate defenses, even if the defense itself is straightforward.
> > > >
> > > > In this paper, we show that our attack overcomes the barrier of gradient aggregation, which is widely believed to be an effective defense and is the **only** security measure deployed in major FL systems. Indeed, the proposed malicious parameters modification attack exhibits identifiable characteristics. However, by bringing our proposed attack to the community's attention, defenders are aware of the new threat and can potentially develop security measures based on the characteristics of model parameters.
> > > >
> > > > Papers on adversarial examples also fit into this scope as they broke an existing defense of “cursory human inspection”. Similarly, backdoor attacks need to be “clean-label” to break an existing defense of crowd-sourced data labeling.
> > > >
> > > >
> > > > *To make an analogy with adversarial examples that we think is more fitting*:
> > > >
> > > > Adversarial examples broke an existing defense of human inspection. However, simple adversarial examples are easily detected by automated inspection. Claiming that our proposed attack has serious weakness because it could be potentially defeated by automated parameter inspection is the same as claiming the initial work on adversarial examples to have serious weakness based on the possibility that simple detectors exist, and potentially not worth publication.
> > > >
> > > > Also subsequently, detectors of adversarial attacks were developed in a series of papers that were only possible, because the initial attacks were published and known to be possible threats to the community. Ultimately though, detection turned out to be a dead-end for adversarial examples: All proposed detectors were able to be broken by adaptive adversarial attacks.
> > > > Looking back to our proposed attack, the relationship between inspection-based defenses and the adaptations an attacker can make also falls into the same situation. We provide detailed discussion of this back-and-forth in our section on mitigations on page 10 of the paper, and this is why we think provable differential privacy will be the better defense against these attacks in the future.
> > > >
> > > > > The [attack] … is not compatible with model training (as the authors have acknowledged this in the response).
> > > >
> > > > We want to reiterate that this is not what we claim.  The proposed attack is completely compatible with training a global FL model. The server simply needs not include the update of the attacked user into the global model. Model training continues as normal during the attack. The server can even re-query the same user with a benign model update in all other rounds, potentially immediately following the attack.

---

> > > > > ### Comment · Reviewer_q4yB · 2022-11-19
> > > > > **The claim that the proposed attack bypasses existing defenses is too strong.**
> > > > >
> > > > > I fail to agree with the author that the proposed attack bypasses existing defenses in two aspects.
> > > > >
> > > > > - **On the analogy to adversarial examples**. In the example of adversarial example detection, the process is that people **first propose adversarial attacks**, and then some **simple detection methods are proposed**, and then they are shown invalid to adaptive attacks, and so on so forth. However, in the case of the proposed attack, it is not exactly the case. Manually inspecting parameter norms (gradient norms) is not a dedicated defense to the proposed attack as far as I know. Instead, it can be a debugging tool to see whether there are gradient explosions. Therefore, this defense is indeed "existing" (although not dedicated to your attack), while adversarial example detectors do not exist before people propose adversarial attacks.
> > > > >
> > > > > - **Only a small amount of DP is required to break the attack**. I checked your response to how the proposed attack reacts with DP. It turns out that with $\sigma = 1e-6$, the proposed attack fails. However, as stated in DLG (https://arxiv.org/pdf/1906.08935.pdf), DLG is able to bypass a DP noise of $1e-4$, and such a scale of noise does not affect model accuracy very much (<1%). While the numbers are not directly applicable to the proposed attack and model architecture, DP is indeed a general-purpose and widely-used defense, and such results cast doubts on the applicability of the attack.
> > > > >
> > > > > I apologize for my misunderstanding about model training and the proposed attack and thank the authors for clarifying it.

---

> > > > > > ### Author Response · Authors · 2022-11-21
> > > > > > **Our attack bypasses defenses implemented in real systems**
> > > > > >
> > > > > > **Our attack is based on real systems:**:
> > > > > >
> > > > > > The attack we discuss bypasses existing defenses as they are described and implemented, see the references we bring up in our work. To be very explicit, here is an [example](https://support.google.com/messages/answer/9327902?hl=en&ref_topic=9326065#zippy=%2Chow-your-data-is-protected) of current FL systems that describes the implemented security measures, which **only includes secure aggregation**.
> > > > > > The attack is further not anomalous in terms of gradient norms, and mechanisms such as gradient clipping do not affect the attack. ´
> > > > > >
> > > > > >
> > > > > > We also want to reiterate our point that knowing the existence of an attack is key to developing corresponding security measures. We completely agree that user devices should audit parameter vectors sent from the server. However, as we discussed above, there’s currently no automated inspection-based defense implemented in real-life FL systems. Furthermore, if a user, or any party in general, were to perform parameter auditing (maybe using existing tools), knowledge of our proposed attack is a prerequisite for the auditing procedure to be effective. After all, if we don’t even know the characteristics of malicious parameter vectors, then what criteria should an auditor use to determine if a user receives a malicious server update or not?
> > > > > >
> > > > > >
> > > > > > **Regarding differential privacy**:
> > > > > >
> > > > > > Directly comparing noise levels across different scenarios is not something we can recommend. Nevertheless, it is possible that DLG is a bit more robust than the attack we propose (this would be a question for a future case study).
> > > > > > Yet, for DLG, gradient noise is not even necessary as a defense, as the attack is defeated by even small levels of gradient aggregation. The attack we describe, on the other hand, is not.
> > > > > >
> > > > > > Overall, we do think that DP is a strong defense against this attack, and this is also mentioned explicitly in our mitigations section. Nevertheless, while a good amount of research has focused on DP recently, industry adoption has not been as widespread so far. We see the value of our paper in incentivizing companies and stakeholders to implement differential privacy.

---

### Official Review · Reviewer_dBzB · 2022-10-31

**Confidence:** 3
**Correctness:** 4
**Technical Novelty And Significance:** 4
**Empirical Novelty And Significance:** 4
**Recommendation:** 6

**Clarity, Quality, Novelty And Reproducibility:**

Clarity: the paper can be presented better.

Quality: the paper is somehow solid.

Novelty: the paper is technically novel.

Reproducibility: the paper provides some experiment details in the appendix. In general, the method should not be very hard to reproduce.


**Strength And Weaknesses:**

Strengths:
- The paper is well-motivated. I agree with the authors’ arguments on the existing approaches and also believe that focusing on only privacy-sensitive data is a practical setting for attackers.
- The attacking setting (e.g., malicious server scenario), background, and related work has been discussed well. Generally, I like the positioning of this paper.
- Although the method is based on a couple of existing works, it is relatively novel.
- The paper conducts extensive experiments to evaluate the attacks with different setups. The results really show the vulnerabilities of federated language model training in a practical large-scale setting.

Weaknesses:
- The paper can be presented better. The method is complicated and section 3.3 is not easy to follow. Since the approach is built based on existing approaches, some details are not included in this paper, which may confuse readers.
- The paper can be written better by discussing possible defense methods and it will also benefit the community and facilitate more research in this direction.
- The paper lacks qualitative attacking examples reconstructed by the approach. It is hard to have a sense about how strong the attack is by only looking at those metrics.
- What is the “total accuracy” metric exactly?


**Summary Of The Paper:**

This paper considers the potential privacy leakage in federated learning settings. The paper argues that previous attacks are too ambitious to extract as much data as possible and this is at the expense of fidelity, so that previous attacks do not work under large-scale settings. The paper proposes an attack on FL which extracts input text that contains targeted privacy-critical words in language modeling training. In the proposed method, the parameters of the transformer are carefully programmed to filter relevant sequences from user data and encode the information in the gradients. The conducted experiments show that the attack works well and outperforms previous approaches in different settings; they show that the attack works even when the batch size is as large as 2000.


**Summary Of The Review:**

In general, I am positive towards accepting the paper. However, I also believe the paper can be better if there is another round of review.

---

> ### Author Response · Authors · 2022-11-18
> **Response to Reviewer dBzB**
>
> We are glad the reviewer finds the paper to be well-motivated, novel, and showing sufficient empirical evidence to support the vulnerabilities of large-scale aggregation. Here we provide detailed responses to the reviewer's individual questions.
>
> **Paper presentation:**
>
> We have **substantially revised** our submission to include more intuition and clarify existing explanations, especially in Section 3. We have added a new Fig.1, showing a qualitative example of how the attack functions and have included a number of summary boxes to guide the reader through the method section.
>  Please also don’t hesitate to provide further questions if there’s still any confusion.
>
> **Discussion of possible defense:**
>
> We believe there are two lines of possible defenses to our proposed attack.
> The first line of defense is based on parameter inspection. Indeed, the malicious modifications of transformer parameters proposed in the paper exhibit identifiable characteristics. For instance, the rows of linear layers contain identical copies of the measurement vector, and a user can inspect the rank of linear layers as a defense. However, an attacker can easily make adaptations to many inspection-based defenses. For example, the attacker can add a small amount of noise to each row of the linear layers, and bypass the rank inspection defense with minimum impact on the attack performance. Therefore, this line of defense naturally puts users on the “back foot” in a back-and-forth between ever-adapting attackers and constantly defending users. Because of this asymmetry, we advise against relying on any one-off set of parameter inspection rules to avoid attacks similar to the one discussed in this paper.
>
>
> Another line of defense evolves around differential privacy. The most common realizations are for users to perform gradient noising/clipping. Here we provide experiments where a user additionally adds Gaussian noise and performs gradient clipping before sending the model update to server. Notice that the presence of noise makes the measurement by Eqn. 7 noisy. Therefore, we only perform embedding recovery using Eqn. 7 when the magnitude of difference between bias gradient is larger than a certain threshold. The experiment is performed on a 3-layer transformer as described in Wang et al. 2021, where we clip gradient norm to 1 and compare performance on different noise scales from {$1e^{-6}$, $1e^{-7}$, $1e^{-8}$, $1e^{-9}$, $1e^{-10}$}, and fix the threshold to be $1e^{-6}$. The experiment also uses batch size of 2 and sequence length of 32, where one of the sequences contains the target word. The experiment results are provided in the following table.
>
>
> | Noise scale| $1e^{-10}$ | $1e^{-9}$ | $1e^{-8}$ | $1e^{-7}$ | $1e^{-6}$ |
> | ---------------| -----------------| ---------------|----------------|---------------|----------------|
> |Total Acc.   | 0.9375  | 0.9375 | 0.9375 | 0.5625 | 0.0313 |
>
> The results show that the attacker is still able to succeed when the noise scale is relatively small, but predictably fails if the gradient becomes sufficiently noisy. Overall, scaling up the magnitude of noise eventually enforces user’s privacy, as promised by theory in differential privacy.
>
> We have also revised the paper to include more in-depth discussion of the possible defense.
>
> **Qualitative examples:**
>
> Here we provide an actual example reconstructed by our attack.
> In this example, the user data contains 64 sequences, each with length of 32, and the transformer architecture is GPT-2. We partially replace one of the sequences starting from the first token with the following sentences:
> > “My credit card number is 1234567. This is a privacy leaking sentence. Can we recover sensitive information?”
>
> Using “credit card number” as a key phrase, here’s the reconstruction given by our attack:
>
> > “**My credit card number** Art **1234567** Fair **This is** 1973 **privacy leaking sentence. Can we recover sensitive information**?“
>
>
> Bold texts represent exact matches on both tokens and positions.
>
> We find here that an attack through panning can target the sequence containing these keywords and reconstruct text that captures sensitive information.
>
> **Meaning of total accuracy:**
>
> For total accuracy, a reconstructed token at a specific position is considered correct only if both the token id itself and the associating position match the ground truth. Intuitively speaking, this metric measures directly how many (sub)-words are leaked with their correct absolute position in a sentence. We’ve added additional clarification on this point in the paper.
>
>
> Ultimately, we are very glad for all the feedback provided. We have revised these points and updated our presentation substantially. We hope you’ll have the chance to take a look into our updated paper, and could think about raising your score based on these changes.

---

### Official Review · Reviewer_7zTS · 2022-10-31

**Confidence:** 3
**Correctness:** 3
**Technical Novelty And Significance:** 3
**Empirical Novelty And Significance:** 3
**Recommendation:** 6

**Clarity, Quality, Novelty And Reproducibility:**

- Novelty: This work presents a novel idea of targeted privacy attack against FL systems. This targeted setting is novel and well-motivated.
- Clarity: This work is well organized. The concepts are either clearly explained or highlighted with proper references.
- Quality: This work has some minor issues. Such as my concerns about the generality of the method as well as the experimental design on FedSGD.
- Reproducibility: I didn’t find the code of this paper. But I still trust the authors regarding their results.


**Strength And Weaknesses:**

#### **Strengths**:
- The targeted attack setting is novel, well-motivated and interesting

#### **Weaknesses**:
- The designed three steps of the proposed attack match the intuition to perform targeted attack, but it seems to me that the proposed techniques are largely inspired by or directly adopted from Fowl et al. (2022), which makes the technical contribution of this work incremental.
- Based on my understanding, this method requires the model gradients returned by the users. Is this the reason that the authors only performed experiments on FedSGD? What if the gradients are not available, how could the proposed attack work (for instance on FedAVG)? This raises my concern on the generality of the proposed method. Maybe the author could provide some discussions and insights on this point as in Fowl et al. (2022) and Fowl et al. (2021).
- In my opinion, the experimental setup is not very sufficient. Firstly, as mentioned above, only FedSGD is studied. Secondly, the selected evaluation metrics seem to be too general and could not really reveal whether the attack is indeed ‘targeted’. In the worst-case, the performance gain could be simply attributed to the additional attention weights added instead of the actually proposed method. It would be more convincing if the author can show that the proposed attack could recover the sequences with **targeted** key words.
- Finally, this might be not so important. But I am curious about whether the proposed attack could surpass very simple defenses? For instance, the users could just add some noises to the gradients when returning the gradients to the server. Will this break this attack? Not to mention more advanced defenses discussed in Fowl et al. (2022).


**Summary Of The Paper:**

This paper proposes a novel privacy attack against FL systems, where the proposed attack extracts text sequences that contain targeted, privacy critical phrases. The targeted attack mechanism is achieved by maliciously modifying the model parameters on the server. In particular, the attack consists of three steps: tagging target sequences, filtering out irrelevant tokens and post-processing. According to the experimental results, the proposed method is more powerful than the selected Deception baseline.

**Summary Of The Review:**

Overall, I think this work is interesting and I rated this work as 6. I do have some concerns and I am not an 100% expert on attacking FL systems, but I am willing to increase my score if the authors could address my concerns above.

---

> ### Author Response · Authors · 2022-11-18
> **Response to Reviewer 7zTS**
>
> We appreciate the reviewer’s recognition that the proposed scenario of targeted text extraction is interesting, novel and well-motivated.  Here we provide detailed responses to the reviewer's individual questions.
>
> **Technical contribution:**
>
> While we do build on a body of previous work for attacking FL systems, we solve a crucial problem with existing attacks - large enough aggregation effectively defeats the attack of Fowl et al. 2022. Especially with more realistic federated architectures, we see the performance of Fowl et al. drop off precipitously. This ultimately puts the attack of Fowl et al. on the same spectrum as honest-but-curious attacks, which all degrade with enough aggregation.
> Because of effects like this, several currently deployed FL pipelines (e.g. google messages) deploy no defense other than secure aggregation. **Our attack fundamentally overcomes this barrier** and shows that these pipelines are unsafe against a server-side attack. Furthermore, there is no easy way to adapt the original implementation of Fowl et al. to industrial-sized batches, and to our knowledge, our novel panning technique is the *first* attack to work against arbitrarily sized batches - additionally a novel empirical contribution.
>
> **How does the attack work on FedAvg:**
>
> In this submission we focus on fedSGD as a template for federated learning. However, this restriction is not as limiting as it might seem: A malicious server can side-step the restriction of FedAVG if the hyperparameters of the FedAVG protocol can be configured so that either the user runs only a single local update step or runs the update on a large enough batch size. Both variants effectively turn FedAVG into FedSGD. We have revised our submission to discuss this option in more detail.
>
>
>
> **Qualitative examples:**
>
> Here we provide an actual example reconstructed by our attack.
> In this example, the user data contains 64 sequences, each with length 32, and the transformer architecture is GPT-2. We partially replace one of the sequences starting from the first token with the following sentences:
> > “My credit card number is 1234567. This is a privacy leaking sentence. Can we recover sensitive information?”
>
> Using “credit card number” as a key phrase, here’s the reconstruction given by our attack:
>
> > “**My credit card number** Art **1234567** Fair **This is** 1973 **privacy leaking sentence. Can we recover sensitive information**?“
>
> Bold texts represent exact matches on both tokens and positions. We find here that an attack through panning can target the sequence containing these key words and reconstruct text that captures sensitive information.
>
>
>
>
> **How does the attack work against simple defense:**
>
> Here we provide experiments where a user additionally adds Gaussian noise and performs gradient clipping before sending the model update to server. Notice that the presence of noise makes the measurement by Eqn. 7 noisy. Therefore, we only perform embedding recovery using Eqn. 7 when the magnitude of difference between bias gradient is larger than a certain threshold. The experiment is performed on a 3-layer transformer as described in Wang et al. 2021, where we clip gradient norm to 1 and compare performance on different noise scales from {$1e^{-6}$, $1e^{-7}$, $1e^{-8}$, $1e^{-9}$, $1e^{-10}$}, and fix the threshold to be $1e^{-6}$. The experiment also uses batch size of 2 and sequence length of 32, where one of the sequences contains the target word. The experiment results are provided in the following table.
>
>
> | Noise scale| $1e^{-10}$ | $1e^{-9}$ | $1e^{-8}$ | $1e^{-7}$ | $1e^{-6}$ |
> | ---------------| -----------------| ---------------|----------------|---------------|----------------|
> |Total Acc.   | 0.9375  | 0.9375 | 0.9375 | 0.5625 | 0.0313 |
>
> **Reproducibility:**
>
> Thank you for bringing this to our attention. We have now added code to the supplementary material to replicate the attack described in this submission.
>
>
> Thank you for all of the provided feedback. We hope to have addressed any question above and in our updated draft. Please let us know if these clarifications in our response and updated draft would make you consider raising your score, or if we should make further changes!

---

### Official Review · Reviewer_8KC8 · 2022-11-02

**Confidence:** 2
**Correctness:** 3
**Technical Novelty And Significance:** 3
**Empirical Novelty And Significance:** 3
**Recommendation:** 6

**Clarity, Quality, Novelty And Reproducibility:**

The writing quality is not great. The paper assumes a lot of familiarity with the transformer architecture and explains things bottom up. Indeed, the reader is facing a lot of technical terms and details without being given the big picture that is crucial.

**Strength And Weaknesses:**

Strength:
I like the aspect of “targeted” extraction of the texts. Here the attacker plants specific traps to extract specific texts from the model. This is a plus for this paper’s attack.

Weakness:
The fact that an attack is done in a malicious setting is a downside. This means that the users can *potentially notice* that the server is *not* following the protocol honestly. The paper has (as far as I can see) zero discussions on this important aspect, and I would like to hear from the authors about this, if possible: Can you argue that the (malicious) changes to the model are indistinguishable for the users?

**Summary Of The Paper:**

This paper is about attacking the privacy of users when they do federated learning in specific settings. The setting is for training (large) language models that specifically use the “transformer” architecture. Moreover, the attack is done in the “malicious” setting, in which the central server who is running the federated learning does *not* behave honestly (& curiously) but it rather sends fabricated models to the users with the goal of extracting personal data from them. The paper is focused on extraction of *targeted* information; e.g., a credit card number as the string that follows the text “credit card”.

The paper compares its findings with recent works that also performs data extraction attacks on federated learning for the same architecture. Since the paper’s focus is on extracting targeted sentences it will do better in doing the exact goal (of this paper) because the other papers aim to extract texts rather indiscriminately.

**Summary Of The Review:**

In summary, I find the targeted aspect of the attacks of this paper interesting and I think this will lead to more papers like this. On the other hand, the paper suffers from a presentation that is not suitable for general audience and does not investigate the limitations of their attacks; e.g., against users who inspect the models sent to them.

---

> ### Author Response · Authors · 2022-11-18
> **Respond to Reviewer 8KC8**
>
> We appreciate the reviewer’s recognition that the proposed scenario of targeted text extraction is interesting.  Here we provide detailed responses to the reviewer’s individual questions.
>
> **Malicious setting and Possible Defenses:**
>
> We agree that the malicious setting is a stronger threat model. At the same time, we have to stress that the malicious server, while breaking the intent of the protocol, still abides by the constraints set out in the FL protocol (i.e. the server only sends a swapped parameter model).
>
> User inspection of the server state would notice the attack as proposed here, as we discuss no mitigations to hide the attack. But, this has two reasons:
> 1. So far, there has been no indication that these attacks are even possible at scale and so no such inspections are implemented currently (see published system designs in Paulik et al., 2021; Dimitriadis et al., 2022). If the attack in this paper is published and FL systems are modified to reliably detect attacks such as this one, we argue this was well worth the effort to publish this vulnerability.
> 2. User devices are generally passive and only follow automated protocols set up by the FL system provider. FL might take place, for example at night (see Hard et al. 2019), when the device is inactive and users are generally not capable of inspecting and understanding model parameters. As such any detection of such an attack *has* to be automated. Detection of exactly the attack that we propose here would be straightforward, but from a broader viewpoint we want to make the point that these transformer models are highly configurable and detection of even a list of known variants of attacks is likely no guarantee of safety against novel variations of the same attack.
> 3. For a practical example, consider a defense that examines the rows of linear layers sent by the attacker to determine if these rows contain identical copies of the measurement vector, and inspects the rank of linear layers as a defense. In response, the attacker can add a small amount of noise to each row of the linear layers, and bypass the rank inspection defense with minimum impact on the attack performance. A larger problem in the cat-and-mouse game between attacker and defense in FL is that both parties have full knowledge of potential defenses implemented on the user side (given that the server has to know how updates are computed/modified on user devices), but attacks can remain hidden until published, giving an advantage to the attacker.
>
> We do think this an important question and have included this discussion in our updated paper.
>
>
> **Writing:**
>
> We have gone through the paper to substantially improve the writing. The proposed attack builds upon previously published work on attacks in FL, but we hope to have made both the bigger picture clearer in our updated draft and made the paper more self-contained.
>
>
> Overall, we are happy for the feedback provided and would be glad to answer further questions or discuss more. We hope that you can find the time to check our revised draft. Please let us know if the updated presentation therein would let you consider raising your score or if there are other things for us to improve.

---

### Official Review · Reviewer_zieL · 2022-11-02

**Confidence:** 3
**Correctness:** 3
**Technical Novelty And Significance:** 3
**Empirical Novelty And Significance:** 3
**Recommendation:** 6

**Clarity, Quality, Novelty And Reproducibility:**

Clarity: The paper is clearly written. However, the approach is quite complicated and some aspects of the approach could be explained in more detail without relying on related work.

Quality: The paper is well written and is mostly easy to follow. Parts where the paper is building on previous work could be explained better. However, the idea is clearly explained and supported by mathematical deductions.

Novelty: The idea of the paper is novel. In previous works the whole input sequence is reconstructed whereas in this paper only parts of the sequence which are relevant to the attacker are reconstructed.

Reproducibility: The experiments seem reasonable. However, no code was shared and a statement that the code will be made public upon acceptance was missing.

**Strength And Weaknesses:**

Strengths:
- The idea of the paper to only reconstruct parts of sequences that are relevant for an attacker is novel.
- This is an important area of research, as the current real-world implementations are based on trustworthiness of the server. Showing that such an attack is possible is showing that changing the implementations and protocols might be necessary to ensure privacy.
- The paper is well written and most of the time easy to follow.

Weaknesses:
- It should be clearer how many heads are needed in total for the attack, as this might be a major limitation. Sudden drastic changes in the architecture when sending the modified model to the users might allow the users to detect this kind of attack.
- It is a bit hard to understand the reasoning behind some of the actions for the attack. For example, what is the reasoning behind shifting the entries to the first d' positions? Without deep knowledge of the related work this paper is building on, it is hard to follow the mathematical reasoning. A short sentence to explain the intuition would be nice and benefit the reader a lot.
- Experimental evaluation is limited. The experiments were conducted only on 2 models Fowl et al. (2022) is for example using another model (Transformer-3) for evaluating their approach.
- The metrics used for evaluating the approach are not explained in detail. This makes it hard to interpret the results.

Questions to Authors:
- Do we need one head for each keyword and additionally one for each of the positional imprinting of each keyword?
- If you write "We fix the sequence length to 32" do you mean that every sequence has the length of exactly 32 tokens or is it possible that sequences have less than 32 tokens which are then padded using padding tokens?

**Summary Of The Paper:**

This paper proposes a new attack on federated learning using text transformers, where the assumption is that the server is not honest-but-curious, but instead malicious and cannot be trusted. Current existing attacks in this scenario try to recover all input tokens of all users. The paper argues that this is not necessary, as an attacker is only interested in specific private information, which in reality is most likely only a small part of the whole sequence. To only reconstruct sequences containing sensitive data, the paper modifies the multi-head attention blocks as well as the linear layers of the transformer and filter for sequences containing specific keywords (e.g., like "credit card" or "social security number").

**Summary Of The Review:**

The paper proposes a novel privacy attack on federated learning using text transformers. The problem statement is well motivated and it is shown that reconstructing only part of sequences that are relevant for the attacker is increasing the performance in contrast to previous approaches. The paper is well written and most of the time easy to follow. If the reader is not very familiar with the work that this paper is building on, it is hard to follow the reasoning.

---

> ### Author Response · Authors · 2022-11-18
> **Respond to Reviewer zieL**
>
> We appreciate the reviewer’s recognition that the proposed attack is novel, the introduced scenario of targeted text extraction is novel, and the paper is well written. Here we provide a detailed response to the reviewer's individual questions.
>
> **How many heads are needed for the proposed attack?**
>
> As described in the paper, exactly one head is needed to imprint a unique sequence identifier. Additionally, one head is required for each individual keyword. The architectures we investigate in our submission all come with 12 heads, which can support targeted text extraction for up to 11 keywords. We believe this is enough for most privacy-critical phrases an attacker might be interested in.
>
> **Reasoning behind shifting the entries to the first $d'$ positions?**
>
> The shifting of entries is in theory not necessary for the construction of our attack. However, performing entries shifting allows us to employ technical tricks to further improve the performance of our attack. In particular, we set the first $d'$ entries of each embedding vector to zero during the construction of the malicious parameters. Consequently, the mixed embeddings after performing malicious self-attention will have their first $d'$ position containing only the signature-related information extracted from entries $d'+1$ to $2d'$ of the target embedding. We observe that this trick improves performance on the implementation level. The alternative here would be to keep these entries unaltered, which would make the unmixing modestly harder.
>
> We have revised the paper to reflect the separation mentioned in this response.
>
> **Experimental evaluation:**
>
> In our submission we evaluate GPT-2 as an example of an autoregressive LM and BERT-base, as an example of a mask-based bidirectional model, as stand-ins for two relevant types of models and training routines. In principle, the attack described in this paper can be launched against any transformer variation of either of these two types, and the effectiveness of the attack would only change based on the overall number of parameters in linear layers and number of heads in the attention. The actual state of the model parameters and minor variations in e.g. layer norm variations, positional embeddings, or additional structure do not affect the attack. We now provide code with our submission to make it possible to evaluate the attack against other model architectures.
>
>
> **Do we need one head for each keyword and additionally one for each of the positional imprinting of each keyword?**
>
> The goal of the positional imprinting module is to encode part of the first token in each sequence to the mixed embedding. For this abstract task only a single head is needed, no matter how many keywords are used or how many sequences are contained in the batch. In the same manner, each additional head can attend to an additional keyword, and do so for any number of sequences in the batch.
>
>
> **Fixing the sequence length to 32:**
>
> As described in the appendix, we construct our user data so each sequence has length of exactly the given sequence length. We construct this data via truncation, so that each sequence is fully filled with data. We note that padding would also be a possibility, but we refrain from doing so, as it would make the attack easier as fewer relevant tokens are included in each sequence.
> Further, a sequence length of 32 is just one the example we consider, we verify that the attack is stable for a variety of sequence lengths in Fig. 9 in the appendix.
>
> **Code Sharing:**
>
> Thank you for bringing this up. We have updated our supplementary material to provide all code required to replicate our experiments.
>
> Thank you for your feedback on our submission. We hope to have clarified all questions raised in your review in our response and our updated draft. We have also substantially improved our presentation to make it more self-contained and clearer to follow. Please check out the updated draft if you can and let us know if there are parts that we should further improve. If not, we would would be happy if you could consider raising your score in response to our revision.

---

### Author Response · Authors · 2022-11-18
**General Comment to ACs and Reviewers**

We thank all the reviewers for their thoughtful feedback. Here we provide a general response to all reviewers and the AC, and provide individual responses to further address the questions of individual reviewers.

We start by stating our appreciation for the reviewers’ recognition. In particular, all reviewers consider the scenario of targeted text extraction proposed in this paper to be novel, well-motivated and practical, and also consider our proposed attack to be relevant to the community.

Based on the reviewer's comments, we provide discussions that address the common questions.

**Qualitative examples:**

Here we provide an actual example reconstructed by our attack.
In this example, the user data contains 64 sequences, each with length of 32, and the transformer architecture is GPT-2. We partially replace one of the sequences starting from the first token with the following sentences:
> “My credit card number is 1234567. This is a privacy leaking sentence. Can we recover sensitive information?”

Using “credit card number” as a key phrase, here’s the reconstruction given by our attack:

> “**My credit card number** Art **1234567** Fair **This is** 1973 **privacy leaking sentence.** **Can we recover sensitive information**?“


Bold texts represent exact matches on both tokens and positions.

We find here that an attack through panning can target the sequence containing these keywords and reconstruct text that captures sensitive information.





**Discussion about inspection-based defense**:

We agree with the reviewers that the malicious modifications of transformer parameters, at least as initially proposed in the paper, exhibit identifiable characteristics. However, there are several problems with relying on parameter inspection as a defense to this attack:
Awareness of our attack is necessary to know what to look for in terms of “conspicuous” parameters - just another reason we urge publication of our work!
Parameter inspection, at least on the server side, is not implemented in any major FL framework (see Bonawitz et al. 2019, Dimitriadis et al. 2022, etc.). This means to get to the point of even being able to inspect parameters, a user would have to “root” their device - making this defense currently infeasible for the average user concerned about privacy.
Such defenses naturally put users on the “back foot” in a back-and-forth between ever-adapting attackers and constantly defending users. As seen in Fowl et al. 2022, an attacker could simply add a small amount of noise to the conspicuous parameters to evade automated defenses, such as ones which consider the rank of different layers.
At the loss level, our modifications would be indistinguishable from a model simply in the early stages of training.

Because of this asymmetry induced by these sorts of defenses, we strongly advise against relying on any one-off set of parameter inspection rules to avoid attacks like this.

**Discussion about defenses based on differential privacy:**

On the other hand, defenses based on differential privacy remain attractive due to their theoretical guarantees. In this category, gradient clipping and noising are the most commonly seen defenses.
We provide experiments where a user additionally adds Gaussian noise and performs gradient clipping before sending the model update to server. Notice that the presence of noise makes the measurement by Eqn. 7 noisy. Therefore, we only perform embedding recovery using Eqn. 7 when the magnitude of difference between bias gradient is larger than a certain threshold. The experiment is performed on a 3-layer transformer as described in Wang et al. 2021, where we clip gradient norm to 1 and compare performance on different noise scales from {$1e^{-6}$, $1e^{-7}$, $1e^{-8}$, $1e^{-9}$, $1e^{-10}$}, and fix the threshold to be $1e^{-6}$. The experiment also uses batch size of 2 and sequence length of 32, where one of the sequences contains the target word. The experiment results are provided in the following table.

| Noise scale| $1e^{-10}$ | $1e^{-9}$ | $1e^{-8}$ | $1e^{-7}$ | $1e^{-6}$ |
| ---------------| -----------------| ---------------|----------------|---------------|----------------|
|Total Acc.   | 0.9375  | 0.9375 | 0.9375 | 0.5625 | 0.0313 |


 The results show that the attacker is still able to succeed when the noise scale is relatively small, but predictably fails if the gradient becomes sufficiently noisy.

Overall, as promised by theory, applying stronger differential privacy eventually enforces the user’s privacy, and we believe related defense algorithms are thus practically attractive. However, it is also known that stronger differential privacy degrades the utility of the final model and finding a good balance between privacy and utility remains an important research direction.

---

> ### Author Response · Authors · 2022-11-18
> **General Response to ACs and Reviewers (Cont.)**
>
> **Discussion about FedAVG:**
>
> In this submission we focus on fedSGD as a template for federated learning. However, this restriction is not as limiting as it might seem: A malicious server can side-step the restriction of FedAVG if the hyperparameters of the FedAVG protocol can be configured so that either the user runs only a single local update step or runs the update on a large enough batch size. Both variants effectively turn FedAVG into FedSGD.
>
> **Paper presentation:**
>
> We appreciate the reviewers' comments that the paper could use a revision to better deliver the content to a general audience. We have **substantially revised** our submission to include more intuition and clarify existing explanations,  especially in Section 3. We have added a new Fig.1 to show a qualitative example of how the attack functions and have included a number of summary boxes to guide the reader through the method section.
>
> We also include a more detailed discussion of possible defenses and updating protocols in the appendix.
>
> Please let us know what you think of the substantially revised presentation and do not hesitate to let us know if there’s any further comments!

---

### Decision · Program_Chairs · 2023-01-20

**Decision:**

Accept: poster

**Justification For Why Not Higher Score:**

The paper clearly has some flaws alluded to by the reviewers so does not merit a higher score.

**Justification For Why Not Lower Score:**

This paper is borderline but I think the setting is new and interesting and more critical work like this needs to be done for federated learning so I lean towards accepting the paper.

**Metareview: Summary, Strengths And Weaknesses:**

In this work, the authors propose the first attack on federated learning (FL) systems that achieves targeted extraction of sequences that contain privacy-critical phrases. They use maliciously modified parameters to allow the transformer to filter relevant sequences from aggregated user data and encode them in the gradient update. The authors claim that their attack can effectively extract sequences of interest even against extremely large-scale aggregation. According to the authors this represents an improvement over current attacks, which are sequence-agnostic and aim to extract as much data as possible from an FL update at the expense of fidelity for any particular sequence. The reviewers thought that the paper presents an attack in a new and more difficult scenario and interesting area of research "as the current real-world implementations are based on trustworthiness of the server".

The reviewers raised some concerns: (1) need more clarity on the number of heads (2) drastic changes in the architecture might make detection of attack easy (3) lack of clear intuition behind some of the logic (4) difficulty in following the mathematical reasoning (5) limited experiments. The authors provided a thorough rebuttal which addressed some of the concerns that were raised. However the reviewers did not change their scores. My own reading of the paper is similar to the reviewers. The paper studies an interesting novel setting but has some flaws. In my view the benefits of the paper and discussing it outweigh its flaws and therefore I recommend acceptance. However, I encourage the authors to address the technical issues raised by the reviewers in the final version of their manuscript.


**Note From Pc:**

if the above contains the word "oral" or "spotlight" please see: "oral" presentation means -> notable-top-5% and "spotlight" means -> notable-top-25%. As stated in our emails, we are disassociating presentation type from AC recommendations

**Summary Of Ac-Reviewer Meeting:**

Due to timezone differences I was not able to schedule a meeting. I did solicit feedback and also read the paper carefully myself. The reviewers do not want to change their scores. Based on the discussion is clear that the paper has a unique setting which makes it interesting. However, the paper also has some shortcomings. The paper is overall borderline but based on the discussion I lean towards accepting the paper because of the novel setting despite its flaws.